# The Curse of CoT: On the Limitations of Chain-of-Thought in In-Context Learning

**Tianshi Zheng**[*1]**, Yixiang Chen**[*1]**, Chengxi Li**[*1]**, Chunyang Li**[1]**, Qing Zong**[1]
**Haochen Shi**[1]**, Baixuan Xu**[1]**, Yangqiu Song**[1]**, Ginny Y. Wong**[2]**, Simon See**[2]
[1]The Hong Kong University of Science and Technology, [2]NVIDIA
{tzhengad, ychenla, clidu}@connect.ust.hk, yqsong@cse.ust.hk

Reviewed on OpenReview: **https://openreview.net/forum?id=7SIrvcYNYj**

## Abstract

Chain-of-Thought (CoT) prompting has been widely recognized for its ability to enhance reasoning capabilities in large language models (LLMs). However, our study reveals a surprising contradiction to this prevailing perspective within the fundamental domain of pattern-based in-context learning (ICL). Through extensive experiments involving 16 state-of-the-art LLMs and nine diverse pattern-based ICL datasets, we demonstrate that CoT and its reasoning variants **consistently underperform** direct answering across varying model scales and benchmark complexities. To systematically investigate this unexpected phenomenon, we designed extensive experiments to validate several hypothetical explanations. Our analysis uncovers a fundamental hybrid mechanism of explicit-implicit reasoning driving CoT's performance in pattern-based ICL: while explicit reasoning falters due to LLMs' struggles to infer underlying patterns from demonstrations, implicit reasoning—disrupted by the increased contextual distance of CoT rationales—often compensates, delivering correct answers despite flawed rationales. This hybrid mechanism explains CoT's relative underperformance, as noise from weak explicit inference undermines the process, even as implicit mechanisms partially salvage outcomes. Notably, even long-CoT reasoning models, which excel in abstract and symbolic reasoning, fail to fully overcome these limitations despite higher computational costs. Our findings challenge existing assumptions regarding the universal efficacy of CoT, yielding novel insights into its limitations and guiding future research toward more nuanced and effective reasoning methodologies for LLMs.

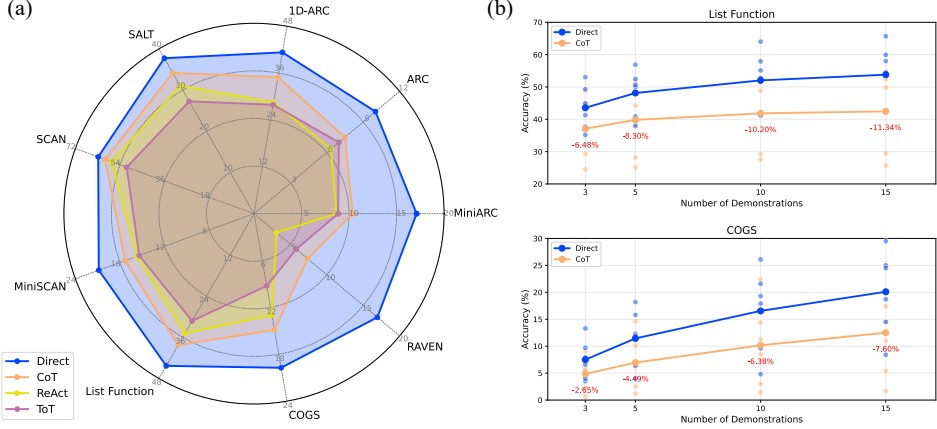

Figure 1: **(a)** Performance of direct answering, CoT, ReAct, and ToT across 9 ICL benchmarks, averaged over 16 LLMs. **(b)** Performance gaps between direct answering and CoT with varying numbers of demonstrations.

---

*Equal Contribution.

# 1 Introduction

Chain-of-Thought (CoT) prompting (Wei et al., 2022) has emerged as a pivotal technique in advancing modern large language models (LLMs). By encouraging models to generate explanatory rationales (i.e., intermediate reasoning steps) prior to producing the final answer, CoT significantly improves the reasoning capabilities of LLMs, enabling them to achieve more accurate and interpretable outcomes. Extensive evidence has demonstrated that CoT is particularly effective in tasks involving mathematical, symbolic, or code-based data, and also leads to substantial improvements in general natural language reasoning and factual reasoning (Sprague et al., 2024; Zheng et al., 2024; Yu et al., 2024). Building upon the foundation of CoT, numerous advanced reasoning frameworks—such as ReAct (Yao et al., 2023b), Tree-of-Thought (ToT) (Yao et al., 2023a), and Graph-of-Thought (GoT) (Besta et al., 2024)—have been proposed to facilitate problem-solving in more sophisticated scenarios. Furthermore, the emerged ability of generating long-CoT reasoning steps has become a driving factor behind advanced reasoning models such as OpenAI o1 (OpenAI, 2024), o3-mini (OpenAI, 2025), and Deepseek-R1 (DeepSeek-AI et al., 2025). Beyond empirical improvements, recent theoretical analyses also indicate that CoT enables transformers to perform inherently serial computations and thus overcome their intrinsic limitations in parallel computation (Li et al., 2024).

Despite the well-established effectiveness of CoT, several studies have also explored its limitations. For instance, Ye & Durrett (2022) conducted experiments on earlier LLMs such as GPT-3 (Brown et al., 2020) and OPT (Zhang et al., 2022), demonstrating that these models may generate unreliable explanations in few-shot textual reasoning scenarios. Additionally, Stechly et al. (2025) highlighted CoT's reliance on problem-specific prompts and its limited scalability in planning tasks. Furthermore, Zhang et al. (2025) showed that although CoT effectively improves performance, it still faces inherent limitations stemming from the complexity involved in navigating the prompt and answer spaces. Nonetheless, CoT remains widely recognized in current LLM literature as a broadly effective approach to LLM problem-solving, consistently outperforming direct answering.

In this paper, we reveal a strikingly counterintuitive finding: Chain-of-Thought prompting unexpectedly degrades LLM performances in certain problem-solving contexts. We investigate in-context learning (ICL) tasks, in which LLMs learn to predict the output of a test instance by extrapolating beyond demonstrations in the form of input-output pairs. Specifically, our analysis focuses on pattern-based ICL benchmarks where the relationships (e.g., patterns, rules, functions) between inputs and outputs are explicitly definable. Through extensive experiments involving 16 modern LLMs and 9 diverse ICL benchmarks (spanning textual, numerical, and symbolic data), we demonstrate that CoT and its reasoning variants (e.g., ToT, ReAct) **consistently underperform** direct answering by a significant margin (Figure 1a). Furthermore, we observe that this performance gap widens as the number of in-context demonstrations increases (Figure 1b). Our findings challenge the prevailing assumption that CoT is universally effective across various reasoning tasks.

To systematically investigate the underlying causes of this unexpected "curse" effect, we formulate and evaluate three core hypotheses through extensive tailored experiments:

- **Hypothesis 1.** The CoT rationale increases the contextual distance between demonstrations and answers, disrupting the few-shot learning structure and thereby degrading performance.

- **Hypothesis 2.** LLMs struggle to infer underlying patterns from demonstrations when using CoT.

- **Hypothesis 3.** LLMs struggle to apply inferred patterns to test instances when using CoT.

The experimental results empirically validate Hypotheses 1 and 2, providing valuable insights into the limitations of Chain-of-Thought prompting in in-context learning scenarios.

Interestingly, we observed that LLMs employing CoT often achieve correct answers even when the inferred patterns are incorrect. This observation points to a **hybrid mechanism in the CoT mechanism for ICL (Hypothesis 4)**: the final prediction arises from an interplay between **explicit** reasoning (articulated through CoT rationales) and **implicit** reasoning (similar to direct answering), where both processes contribute to pattern inference and execution. However, LLMs' limited ability to infer accurate patterns explicitly

(as validated by Hypothesis 2) introduces noise into the reasoning process, as flawed rationales disrupt the prediction pipeline. Compounding this issue, the increased contextual distance caused by CoT's inserted rationales further diminishes the efficacy of implicit reasoning (as validated by Hypothesis 1). Consequently, CoT prompting underperforms direct answering, which relies exclusively on robust implicit mechanisms. Further experiments reveal that even long-CoT reasoning models—despite consuming **40×** more inference tokens—achieve only comparable or inferior performance to standard LLMs using direct answering.

In summary, our findings advocate for a more nuanced perspective on Chain-of-Thought prompting. Although CoT has demonstrated considerable success in enhancing the reasoning capabilities of large language models, our analysis has revealed critical limitations, especially within pattern-based, in-context learning scenarios. By providing deeper insights into the underlying mechanisms behind these limitations, we highlight that the benefits of CoT rationales are not universally applicable, emphasizing the need for adaptive and context-aware reasoning approaches. Consequently, this work contributes to a more balanced and comprehensive understanding of CoT, informing the development of more robust and flexible reasoning methodologies, and paving the way for future innovations aimed at optimizing large language model performance.

## 2 Preliminaries

Our investigation focuses on **in-context learning tasks characterized by explicitly defined input-output functions**. Specifically, a consistent and verbalized pattern governs the relationship between each input-output pair within the demonstrations. In this section, we provide a formal definition of pattern-based in-context learning and describe model inference under both direct answering and Chain-of-Thought prompting.

### 2.1 Pattern-based in-context learning

In pattern-based in-context learning, LLMs are provided with a limited number of demonstration pairs, each comprising an input and its corresponding output. These pairs adhere to an explicit, consistent, and verbalizable pattern or rule. Formally, the task can be defined as follows:

Given a set of demonstration examples $\mathcal{D} = \{(x_1, y_1), (x_2, y_2), \ldots, (x_k, y_k)\}$, where each input-output pair $(x_i, y_i)$ conforms to a specific pattern or rule $f$, the goal is to predict the output $y_{test}$ for a new input $x_{test}$, where $(x_{test}, y_{test})$ also adheres to the same underlying pattern $f$. Formally, we have:

$$y_i = f(x_i) \quad \text{for all } (x_i, y_i) \in \mathcal{D} \cup \{(x_{test}, y_{test})\}.$$

The pattern-based ICL tasks examined in this paper span various types of data, including textual, numerical, and symbolic data, and involve explicit rules such as arithmetic progressions, logical relationships, string manipulations, or symbolic transformations. Pattern-based ICL is best understood from a task of **pattern induction**, where the model identifies and applies a unified rule from a set of input-output pairs. Therefore, further tasks such as complex mathematical reasoning requiring explicit multi-step solutions is not within our scope, though such tasks may be discussed in the broader context of general ICL.

### 2.2 Direct answering vs. chain-of-thought prompting

In this subsection, we define and compare two prompting paradigms central to our analysis—Direct Answering and Chain-of-Thought Prompting.

**Direct Answering** In the Direct Answering paradigm, the LLM generates the test output $y_{\text{test}}$ based solely on the provided instructions, in-context demonstration examples $\mathcal{D}$, and the test input $x_{\text{test}}$. Formally, the problem-solving process can be modeled as:

$$p(y_{\text{test}} \mid x_{\text{test}}, \mathcal{D}, \text{Instructions})$$

Here, the model is explicitly required to produce the final output directly, without generating intermediate reasoning steps or explanatory rationales.

Table 1: In-context learning datasets in our experiments.

| Dataset | # Demos | Modality | Size |
|---------|---------|----------|------|
| ARC-AGI (Chollet, 2019) | 2~10 | Symbolic | 835 |
| MiniARC (Kim et al., 2022) | 2~8 | Symbolic | 149 |
| 1D-ARC (Xu et al., 2024) | 3 | Symbolic | 901 |
| SCAN (Lake & Baroni, 2018) | 5~8 | Textual | 1,000 |
| MiniSCAN (Nye et al., 2020) | 14 | Textual | 1,000 |
| COGS (Kim & Linzen, 2020) | 10 | Textual | 1,000 |
| SALT (Zheng et al., 2025a) | 4 | Textual | 1,200 |
| List Function (Rule, 2020) | 3 | Numerical | 1,250 |
| RAVEN (Zhang et al., 2019) | 2 | Numerical / Symbolic | 1,259 |
| **Total** | | | **8,594** |

**Chain-of-Thought Prompting** In contrast, Chain-of-Thought Prompting involves a two-stage response process. First, the LLM generates explicit intermediate reasoning or rationale conditioned on the instructions, demonstrations $\mathcal{D}$, and test input $x_{\text{test}}$. Second, it produces the final output $y_{\text{test}}$ based on this rationale, alongside the original context (instructions, $\mathcal{D}$, and $x_{\text{test}}$). This process is formally expressed as:

$$p(\text{rationale} \mid x_{\text{test}}, \mathcal{D}, \text{Instructions}) \cdot p(y_{\text{test}} \mid \text{rationale}, x_{\text{test}}, \mathcal{D}, \text{Instructions})$$

Notably, the demonstration examples $\mathcal{D}$ are identical in both paradigms and do not include explicit reasoning steps. Consequently, our targeted task formulation differs from the few-shot CoT approaches commonly employed in standard QA tasks, where demonstrations explicitly illustrating CoT reasoning steps are provided. Additionally, we experimented with advanced reasoning frameworks, including ReAct and Tree-of-Thought prompting, in which explicit reasoning guidance is provided prior to the task instructions. The detailed prompting template is presented in Appendix D.

## 3 Datasets and models

**Datasets** We conduct experiments on a diverse selection of pattern-based in-context learning datasets spanning multiple modalities[1]: 1) **Symbolic**: Pattern-based transformations between symbolic matrices, e.g., ARC-AGI and MiniARC. 2) **Textual**: Rule-based translations between natural language and artificial languages, e.g., SCAN and COGS. 3) **Numerical**: Pattern-based or function-based projections between numerical vectors or matrices, e.g., List Functions and RAVEN.

All selected datasets are reasoning-intensive, stress-testing the abstract and inductive reasoning abilities of LLMs. Details of datasets are provided in Table 1. We include further data processing details in Appendix C.

**Models** We evaluated 16 open-source and proprietary LLMs with varying parameter sizes, with details in Appendix A. Note that long-CoT reasoning models, such as o1 and Deepseek-R1, are analyzed separately from the main group of LLMs due to their limited compatibility with direct answering. As these models are specifically optimized for multi-step explicit reasoning, they provide a crucial test case for our hypothesis and are therefore given a dedicated analysis in Section 6.

## 4 Main results

The main experimental results are illustrated in Figure 2 (full results in Appendix E). Across nine ICL benchmarks, LLMs employing **direct answering substantially outperform CoT**, achieving a relative improvement of **20.42%** (absolute 5.10%). Compared to ReAct and ToT, direct answering yields relative improvements of **36.34%** and **47.17%** (absolute 8.02% and 9.64%), respectively. In terms of task modality,

---

[1] https://github.com/HKUST-KnowComp/CoT-ICL-Eval

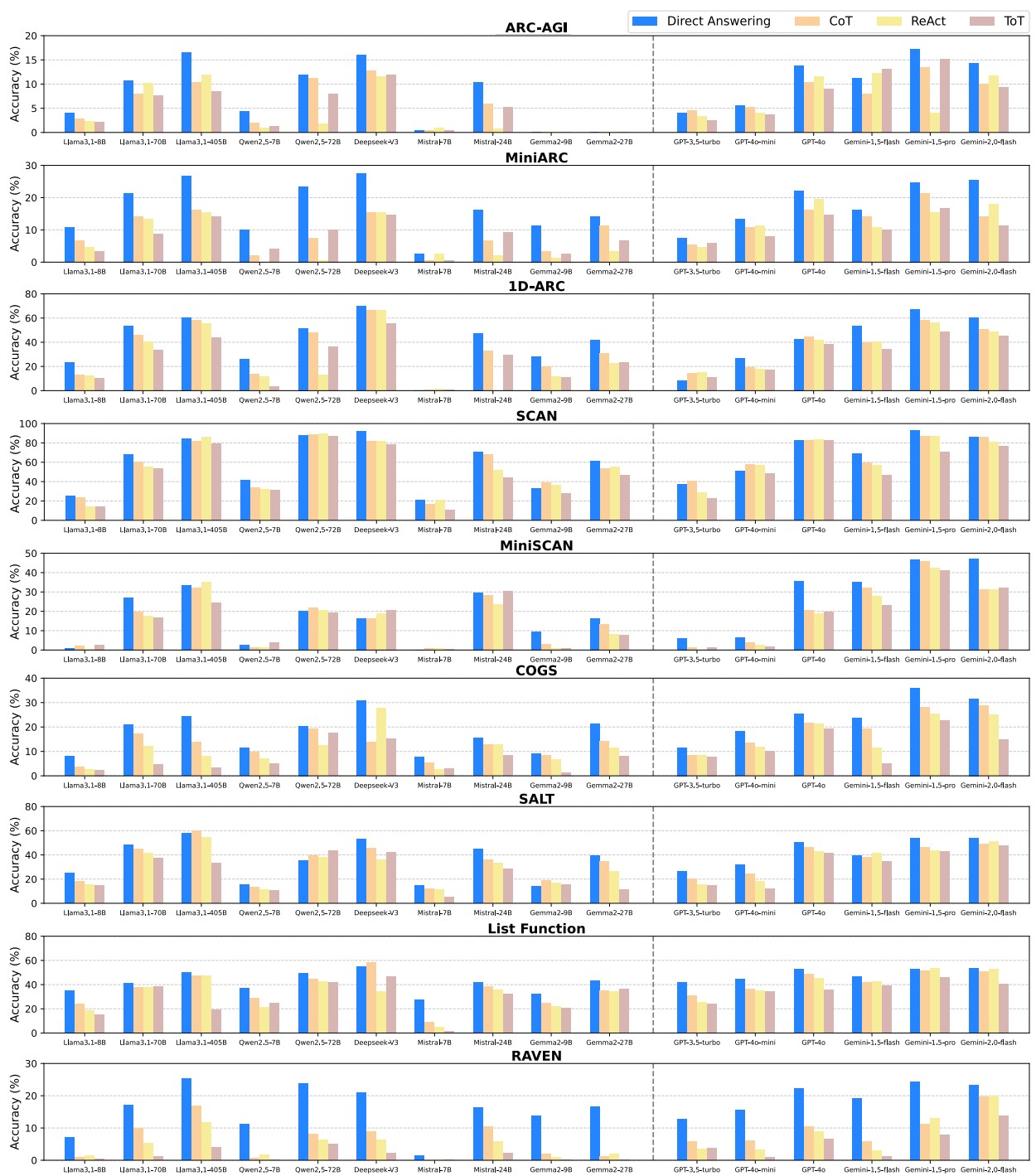

Figure 2: Detailed benchmark performance of LLMs with direct answering, CoT, ReAct, and ToT. Gemma2 models were excluded from ARC-AGI experiments due to limited context length.

the performance gap between direct answering and CoT is **most significant on symbolic ICL tasks** (i.e., ARC-AGI, MiniARC, 1D-ARC, and RAVEN), with a relative improvement of 41.88%; in contrast, this advantage decreases to 10.42% on textual ICL tasks (i.e., SCAN, MiniSCAN, COGS, and SALT).

Regarding model size, since most benchmarks used in our study can be considered relatively out-of-distribution

compared to the LLM training corpora[2], smaller LLMs (e.g., Llama3.1-8B, Qwen2.5-7B) tend to exhibit lower overall performance, as well as more pronounced limitations when utilizing CoT or other reasoning variants. In contrast, larger models (e.g., GPT-4o, Deepseek-V3) achieved better overall performances, in which reasoning frameworks occasionally achieve performance comparable to direct answering. Nevertheless, there are only a few entries in which reasoning frameworks yield a positive outcome from the additional consumption of inference tokens.

Moreover, for two benchmarks that allow flexibility in the number of demonstrations (COGS and List Function), we conduct experiments by varying the demonstration count in the context, ranging from 3 to 15. As illustrated in Figure 1b, the performance gap between direct answering and CoT **widens as the number of shots increases**. This further substantiates the limitations of CoT under different contextual configurations.

These experimental findings reveal a surprising "curse" of CoT, where reasoning frameworks consistently underperform direct answering in pattern-based ICL tasks—with more sophisticated variants (ReAct, ToT) performing even worse. This counterintuitive phenomenon challenges conventional assumptions about the benefits of explicit reasoning in LLMs and motivates our systematic investigation into the underlying mechanisms behind this performance degradation.

## 5    Why chain-of-thought fails in in-context learning?

In this section, we systematically diagnose the root causes of CoT's inefficacy through a hypothesize-and-test methodology. We design targeted experiments to validate or refute potential explanations for this limitation. Details of all four experiments are in Appendix B.

### 5.1    The Contextual Distance Curse: how CoT disrupts few-shot learning

In-context learning, as delineated by Brown et al. (2020), assumes that few-shot demonstrations are presented as a coherent, uninterrupted sequence, enabling the model to process them as a unified contextual signal for learning. However, under Chain-of-Thought prompting, the insertion of intermediate rationales between demonstrations and the final answer prediction may disrupt this continuity. We thus propose our first hypothesis:

**Hypothesis 1.** The CoT rationale increases the contextual distance between demonstrations and answers, disrupting the few-shot learning structure and thereby degrading performance.

Formally, we define ***contextual distance*** as the number of tokens separating the end of the in-context demonstrations (i.e., the input-output pairs provided in the prompt) from the position in the sequence where the model begins generating the final answer output. In the context of Chain-of-Thought prompting, this distance generally equals the token length of the generated reasoning rationale, which is inserted between the demonstrations and the answer.

To test this hypothesis, we designed two controlled experiments to isolate and evaluate the effect of contextual distance and CoT:

**Dummy Rationale Experiment**   To disentangle the semantic content of CoT from its structural impact, we instructed LLMs to generate a semantically neutral "dummy" rationale prior to predicting the final answer, thereby preserving the contextual distance while eliminating reasoning-specific effects. We controlled two variables: modality and length. For modality, we considered textual and symbolic data. In the textual condition, LLMs recited excerpts from Shakespeare's Sonnets; in the symbolic condition, they generated a countdown list from a specified integer to one. These tasks were chosen to minimize generation variance and prevent unbounded outputs. For length, we varied the dummy rationale size: reciting 1, 2, 4, or 8 sonnets (approximately 150 tokens per sonnet) and counting down from 50, 100, 200, or 400 (approximately 3 tokens per number). This yielded contextual distances ranging from 150 to 1200 tokens, encompassing typical CoT rationale lengths (150 to 500 tokens).

---

[2]The SCAN dataset might be subject to data contamination to some extent.

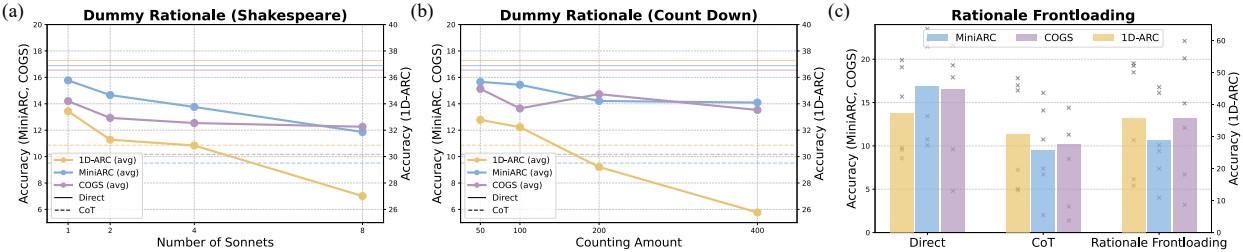

Figure 3: **(a)** Average performance with dummy rationale in Shakespeare's Sonnet. **(b)** Average performance with dummy rationale in countdown list. **(c)** Effect of rationale frontloading. All scores represent mean accuracies across six LLMs.

It is important to acknowledge that this "dummy rationale" approach is not without its limitations. Producing Shakespeare-like text or a procedural countdown is not identical to producing a task-relevant rationale and may introduce process differences unrelated to contextual distance (e.g., stylistic or format-specific generation effects), which is distinct from the structural effect of contextual distance. Despite these potential confounders, these tasks were chosen for their highly controllable output length and low semantic overlap with the pattern-recognition problems, allowing us to introduce a lengthy, unrelated text block that structurally mimics a CoT rationale. Therefore, while not a perfect control, the consistent performance degradation observed in this experiment provides strong evidence that is highly consistent with our contextual distance hypothesis.

Note that we adopt **in-response** dummy rationales, as this approach preserves the relative positioning between the chat template and the rationale, consistent with standard CoT prompting. However, an alternative format that involving **in-prompt** dummy rationales has also been proposed (Lanham et al., 2023). Experimental results demonstrate that both methods provide evidence supporting the contextual distance curse (see Appendix B.1.2).

**Rationale Frontloading Experiment**   To preserve CoT semantics while eliminating contextual distance, we first elicited CoT rationales from LLMs for each test instance. Specifically, for each instance, we obtained its rationale under the regular CoT setting, then reused the same rationale (with the final answer removed to prevent leakage) in the front-loading condition by placing it before the demonstrations and query. Thus, the rationale content is identical across conditions, and only its position in the prompt (i.e., the contextual distance) differs. This approach ensures that the model has access to the same reasoning content, without separating demonstrations from the answer prediction.

The experimental results are presented in Figure 3. From the dummy rationale experiment, we observe that LLM performance generally declines as contextual distance increases. The only exception occurs in the countdown task on the COGS dataset, where LLMs frequently refuse to generate dummy rationales when instructed to count down from 200 or 400. From the rationale frontloading experiment, we find that performance substantially improves when rationales are prepended to the in-context demonstrations. These results provide significant evidence supporting Hypothesis 1. However, we also note that dummy rationales outperform CoT (on MiniARC and COGS), even at greater contextual distances, while frontloaded rationales still underperform relative to direct answering. These observations suggest that contextual distance alone does not fully account for the observed "curse". Additional limitations inherent to CoT itself must also contribute to its inefficacy.

**Findings:** Hypothesis 1 is validated; however, it does not fully explain the CoT curse.

## 5.2   Pattern inference vs. execution: two stages of failure

Chain-of-Thought in pattern-based in-context learning is commonly regarded as a two-stage process: first, LLMs infer the underlying pattern or rule from the provided demonstration pairs, and second, they apply this

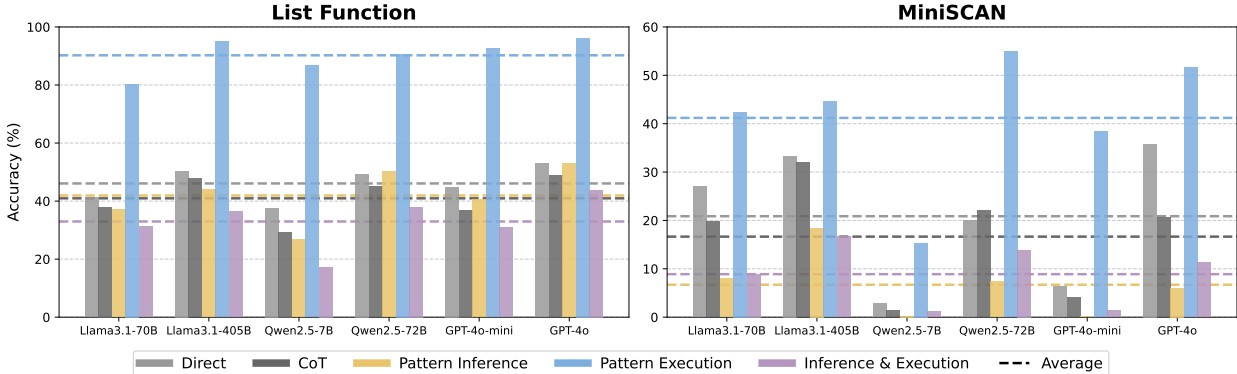

Figure 4: Performance comparison of pattern inference and execution across two benchmarks (List Function and MiniSCAN) and six LLMs.

inferred pattern to generate predictions for test instances (Liu et al., 2024; Zheng et al., 2025a). Given the observed deficiencies of CoT in our experiments, we propose two hypotheses to dissect the potential sources of this failure:

**Hypothesis 2.** LLMs struggle to infer underlying patterns from demonstrations when using CoT.

**Hypothesis 3.** LLMs struggle to apply inferred patterns to test instances when using CoT.

To rigorously test these hypotheses, we designed a two-phase experiment to independently evaluate LLM performance across both stages: pattern inference and pattern execution. For this analysis, we selected two datasets—List-Function and MiniSCAN—which allow for precise evaluation against ground-truth patterns. In the pattern inference stage, we assessed whether LLMs could correctly infer the underlying pattern (e.g., a Python function or symbolic rule) from input-output pairs in demonstrations. In the pattern execution stage, we evaluated their ability to apply the ground-truth pattern to test instances. We further evaluated "Inference & Execution" performance by collecting the inferred pattern (without format constraints) from CoT reasoning and directly providing it for pattern execution. This setup aims to further validate the consistency between the original CoT process and a separate inference-then-execution mechanism.

The experimental results are depicted in Figure 4. Across both datasets, we observe that LLM performance in pattern inference consistently falls below that of pattern execution. This disparity suggests that the primary challenge for LLMs using CoT lies in accurately deducing the underlying rules from demonstration pairs. In contrast, their ability to execute a given pattern appears relatively stronger, though still imperfect.

On the other hand, we also observe that the overall performance of "Inference & Execution" remains significantly lower than that of the original CoT across both datasets. This pronounced disparity—combined with the observations above—suggests that CoT often produces correct answers despite incorrect pattern inference in numerous cases (we provide further case studies in Appendix B.4 for an intuitive demonstration). In other words, rather than relying solely on explicit pattern execution for the test instance, the LLM's answer generation under CoT appears to be implicitly influenced by the in-context demonstrations, which improves the final performance. This interpretation is supported by experimental results in Appendix B.5.

This challenges the simplistic assumption of CoT as a strictly two-stage mechanism. Instead, these results suggest the presence of implicit reasoning mechanisms within CoT, whereby LLMs leverage latent pattern recognition and execution—akin to the processes underlying direct answering—to compensate for shortcomings in explicit inference and execution.

**Findings:** Our results indicate that pattern inference (H2) constitutes a more significant bottleneck for LLMs than pattern execution (H3): the primary failure point lies in inducing the correct rule from demonstrations. Nonetheless, our evidence also suggests that implicit mechanisms beyond explicit rule-following may also contribute substantially to CoT performance.

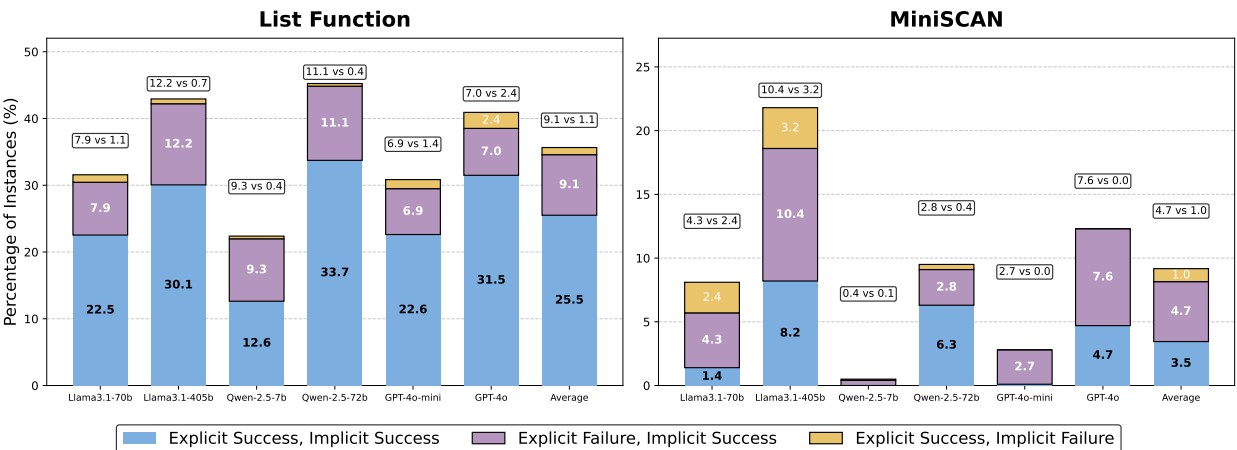

Figure 5: Decomposition of CoT success: contributions from explicit and implicit reasoning.

### 5.3 The Explicit-Implicit Hybrid Mechanism: uncovering divergent answers and rationales

Findings from the preceding analyses provide compelling empirical evidence for a new conceptualization of CoT in in-context learning: the **Explicit-Implicit Hybrid Mechanism**. This perspective posits that the final prediction in CoT emerges from a composite process involving both explicit pattern inference and execution (articulated through the CoT rationale) and implicit pattern recognition and execution (latent reasoning akin to direct answering). The observed discrepancy between the poor performance of explicit pattern inference (Section 5.2) and the relatively high accuracy of CoT predictions suggests that implicit mechanisms may compensate for deficiencies in the explicit reasoning process. Building on this insight, we formulate and investigate a hypothesis that characterizes this dual process:

**Hypothesis 4.** In pattern-based ICL, CoT predictions can be modeled as arising from a dual process of explicit and implicit reasoning. This process appears to be asymmetric: implicit reasoning is the dominant driver of successful predictions, compensating for the frequent ineffectiveness of explicit reasoning.

It is challenging to precisely disentangle the effects of implicit and explicit reasoning within a single CoT process. To investigate their relative contributions, we adopt an analytical framework that relies on operationalizing these concepts with proxies.

**Analytical Premises:** First, we treat **direct answering** (i.e., prediction without an intermediate rationale) as a functional proxy for a purely *implicit reasoning* process. While we acknowledge that direct answering may involve its own un-verbalized reasoning, it serves as a baseline for the model's performance without explicit rationalization. Second, we treat correct **pattern inference**[3] from demonstrations as a proxy for successful *explicit reasoning*. We then analyze instances where the LLM with CoT solved a given problem $i$ successfully, attributing the success as follows:

1. **Implicit Contribution:** We attribute the success on instance $i$ to implicit reasoning if the model also solves $i$ using direct answering but fails to infer the correct pattern from the demonstrations (i.e., explicit reasoning fails).

2. **Explicit Contribution:** Conversely, we attribute the success to explicit reasoning if the model correctly infers the pattern but fails to solve $i$ using direct answering (i.e., implicit reasoning fails).

The results of this analysis are illustrated in Figure 5, where instances of implicit-only and explicit-only contributions are shown in purple and yellow, respectively. Across both datasets, the percentage of cases where implicit reasoning drives CoT success is substantially higher—by **7.5×** on List Function and **3.6×**

---

[3]Using "pattern inference" alone provides a conservative upper bound for the contribution of explicit reasoning, thereby strengthening our conclusion.

Table 2: Performance comparison between direct answering of LLMs and long-CoT LRMs. *Token cost represents the weighted sum of context and inference tokens with a 0.25:1 ratio.

| Models | | MiniARC | COGS | RAVEN | Average | |
|---|---|---|---|---|---|---|
| | | | | | Accuracy (%) | Token Cost* |
| LLM (direct) | Qwen2.5-32B | 24.16 | 21.80 | 12.07 | 19.34 | 207.44 |
| | Qwen2.5-72B | 23.49 | 20.40 | 23.67 | 22.52 | 198.54 |
| | Gemini-1.5-pro | 24.83 | **36.00** | 24.31 | **28.38** | 198.95 |
| | Llama-3.1-405B | 26.71 | 24.40 | 25.34 | 25.48 | 201.61 |
| | Deepseek-V3 | 27.52 | 30.80 | 21.05 | 26.46 | 189.71 |
| LRM (long-CoT) | QwQ-32B | 18.70 | 13.00 | 8.82 | 13.51 | 1736.91 |
| | o1-mini | **30.20** | 10.60 | 15.25 | 18.68 | 3072.02 |
| | Deepseek-R1 | 28.86 | 24.00 | **27.56** | 26.81 | 2432.36 |

on MiniSCAN—than the converse scenario. This disparity underscores the dominance of implicit reasoning, which frequently compensates for flawed explicit pattern inference to achieve a correct final answer.

**Findings:** Implicit reasoning significantly outweighs explicit reasoning in contributing to CoT success, validating the asymmetric hybrid mechanism proposed in Hypothesis 4.

**Summary:** Our analysis suggests a compelling explanation for CoT's underperformance in this domain: it operates as a dual-process mechanism, combining both explicit and implicit reasoning (Hypothesis 4), but this mechanism is fundamentally compromised. First, its explicit reasoning pathway appears unreliable, primarily because LLMs struggle to correctly infer underlying patterns from demonstrations (Hypothesis 2). Second, the very structure of CoT—the insertion of a rationale—increases contextual distance, which degrades the performance of its implicit reasoning pathway (Hypothesis 1), a pathway that otherwise appears robust. With its explicit component proving ineffective and its implicit component hampered, CoT consistently underperforms direct answering, a method that relies solely on an uncompromised implicit process.

# 6 The case of specialized reasoning models

A crucial test of our hypothesis involves Large Reasoning Models (LRMs), which are explicitly designed to excel at complex, multi-step reasoning. In pattern-based ICL, a key distinction between these LRMs and traditional LLMs lies in the former's ability to iteratively propose and refine hypothesized patterns, thereby enhancing explicit pattern inference capabilities. However, this extended reasoning process significantly increases contextual distance, creating a direct trade-off with the implicit reasoning pathway.

To evaluate this trade-off, we conducted experiments using three LRMs across three benchmarks, each selected from a distinct task modality. As shown in Table 2, the results are striking: top-performing LLMs using direct answering achieve **comparable or superior performance** to the specialized LRMs. This outcome is particularly notable given that the LRMs consume, on average, **12× more total tokens and 40× more inference tokens**.

These findings suggest that even when the explicit reasoning component is significantly enhanced, its benefits are insufficient to overcome the structural disadvantages imposed by the CoT framework in these tasks. The performance of the implicit pathway, hampered by increased contextual distance, remains a critical bottleneck. This underscores the severity of the "CoT curse" in this domain and highlights the need for strategies that can integrate verbalized and latent reasoning more efficiently.

# 7 Related Work

**Inductive Reasoning** Our pattern-based in-context learning setup aligns with inductive reasoning: models must infer an implicit pattern from demonstration pairs and apply it to a test instance. Inductive reasoning capabilities in large language models have been extensively studied, with a focus on the design of reasoning

frameworks and training or prompting strategies that elicit such behavior (Wang et al., 2024; Qiu et al., 2024; Liu et al., 2024; Zheng et al., 2025a; Li et al., 2025). These capabilities play a critical role across diverse application domains, including financial modeling (Goel et al., 2025; Stempień & Ślepaczuk, 2025) and scientific discovery (Shojaee et al., 2025; Zheng et al., 2025b;c).

**Effectiveness of Chain-of-Thought Prompting**   Chain-of-Thought prompting (Wei et al., 2022) is widely recognized for improving reasoning performance. At the same time, evidence suggests that CoT offers limited gains for tasks that are less reasoning-intensive, such as factuality calibration (Zong et al., 2025b;a; Liu et al., 2025) and semantic classification (Sprague et al., 2024). Recent work has systematically examined how CoT length affects performance (Jin et al., 2024; Wu et al., 2025; Nayab et al., 2025). In contrast, our work provides the first evidence, to our knowledge, that CoT exhibits substantial limitations on a reasoning-intensive task setting where success depends on inferring and generalizing latent patterns from demonstrations.

## 8   Conclusion

In this work, we identify and rigorously analyze a fundamental paradox in Chain-of-Thought prompting: despite its success in various reasoning tasks, CoT consistently underperforms direct answering in pattern-based in-context learning—a fundamental task that harnesses the inductive and abstract reasoning capabilities of LLMs. Our investigation reveals that this failure stems from a flawed **explicit-implicit hybrid mechanism** at the core of CoT's mechanism. We find that the explicit reasoning pathway, responsible for articulating rationales, is unreliable, as large language models struggle to correctly infer underlying patterns from demonstrations. Simultaneously, the very structure of CoT—the insertion of these rationales—increases contextual distance, degrading the performance of the otherwise robust implicit reasoning pathway that powers direct answering. With its explicit component proving ineffective and its implicit component hampered, CoT is fundamentally compromised in these tasks.

Our analysis of specialized Large Reasoning Models further validates this conclusion, demonstrating that even architectures designed for superior explicit reasoning fail to overcome these inherent limitations, performing on par with or worse than direct answering despite incurring substantial computational overhead. These findings challenge the presumed universal efficacy of CoT, highlighting the critical need for a more nuanced understanding that balances explicit and implicit processes. We advocate for the development of adaptive reasoning strategies that can harness the strengths of both modes, paving the way for more robust and efficient large language models.

## Acknowledgement

We thank all the anonymous reviewers and editor for their valuable comments. The authors of this paper were supported by the ITSP Platform Research Project (ITS/189/23FP) from ITC of Hong Kong, SAR, China, and the AoE (AoE/E-601/24-N), the RIF (R6021-20) and the GRF (16205322) from RGC of Hong Kong, SAR, China. We also thank the support from NVIDIA AI Technology Center (NVAITC).

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

# A   Model details

In our experiment, we tested 20 modern LLM/LRMs. All experiments with temperature set to zero.

- **Deepseek-V3 (671B)** (DeepSeek-AI et al., 2024) is a state-of-the-art open-source LLM released by Deepseek.

- **Deepseek-R1 (671B)** (DeepSeek-AI et al., 2025) is a leading open-source LRM trained with reinforcement learning using a rule-based reward system.

- **Gemma-2-9B / Gemma-2-27B** (Gemma Team et al., 2024) is an open-source, lightweight yet high-performance LLM series.

- **Llama-3.1-8B / Llama-3.1-70B / Llama-3.1-405B** (Meta, 2024) is an open-source dense model series incorporating Direct Preference Optimization (DPO) (Rafailov et al., 2024) for alignment.

- **Mistral-7B Instruct v0.3** (Jiang et al., 2023) is an early high-performance and lightweight open-source LLM.

- **Mistral Small 3 (24B)** (Mistral AI, 2025) is the latest high-performance open-source LLM from Mistral, designed for efficiency.

- **Qwen-2.5-7B / Qwen-2.5-32B / Qwen-2.5-72B** (Qwen et al., 2025) is an open-source MoE LLM series pre-trained on 18 trillion tokens and fine-tuned with 1 million examples.

- **QwQ-32B** (Qwen Team, 2025) is a reasoning-focused LLM trained via reinforcement learning, achieving competitive performance through scalable RL and agent-integrated reasoning for tool use and environmental adaptation.

- **Gemini-1.5-flash / Gemini-1.5-pro** (Google, 2024) is a proprietary MoE LLM series optimized for processing ultra-long sequences.

- **Gemini-2.0-flash** (DeepMind, 2024) is the latest proprietary LLM in the Gemini series, featuring enhanced multimodal understanding and reasoning capabilities.

- **GPT-3.5-turbo** (OpenAI, 2022) is a proprietary conversational LLM fine-tuned via RLHF (Ouyang et al., 2022) and PPO (Schulman et al., 2017).

- **GPT-4o-mini / GPT-4o** (OpenAI, 2024) is a proprietary multimodal LLM from OpenAI with enhanced reasoning capabilities.

- **o1-mini** (OpenAI, 2024) is a proprietary LRM from OpenAI utilizing reinforcement learning for inference-time scaling.

# B  Experiment details

We here provide detailed information of our four tailored experiment to investigate the underlying cause of CoT's ineffectiveness in ICL.

## B.1  Dummy rationale experiment

### B.1.1  Prompt instruction

We aim to have LLMs output dummy rationales under controlled modalities to isolate the semantic content of CoT while maintaining contextual distance. The main challenge lies in controlling LLMs to produce outputs of a specific length in symbols or text while minimizing content variance and preventing unbounded outputs. To address this, we instructed LLMs to generate a countdown list from a specified value to one or to recite selected excerpts from Shakespeare's sonnets. In the Shakespeare dummy rationale generation, LLMs occasionally produced minor errors in precise wording, but outputs were consistently controlled to exactly 14 lines (approximately 150 tokens). In contrast, during countdown list generation, some "smarter" LLMs refused to produce the full list when the starting number exceeded 200. This behavior occurred only with the textual modality dataset (COGS); for the other two symbolic/numerical datasets, the generation performed well. Overall, these minor divergences in both experiments did not impact the experimental findings, which indicate that an increase in contextual distance reliably degrades ICL performance.

The prompt instructions for our dummy rationale experiments are provided below:

---

**Prompt Templates**

**Shakespeare**

```
<regular question instructions and data>

Before generating your answer, you must first recite the first n sonnet(s) of Shakespeare's sonnets.

Your output should strictly follow the json dict format below:
{
    "recitation": "your recitation",
    "answer": "your answer"
}
```

---

**Count Down**

```
<regular question instructions and data>

Before generating your answer, you must first count down from n to 1 ([n, n-1, ..., 1]).

Your output should strictly follow the json dict format below:

{
    "countdown": your countdown list,
    "answer": "your answer"
}
```

---

**B.1.2 In-prompt vs. In-response dummy rationale**

In our dummy rationale experiment (Section 5.1), we primarily adopt in-response dummy rationales, as this approach preserves the relative positioning between the chat template and the rationale, consistent with standard CoT prompting. However, an alternative format—involving in-prompt dummy rationales—has also been proposed (Lanham et al., 2023). To evaluate the robustness of our findings, we conducted additional experiments comparing these two formats on the MiniARC and COGS datasets. As shown in Tables 3 and 4, both in-response and in-prompt approaches yield similar performance trends: LLM accuracy generally declines as the contextual distance (i.e., rationale length) increases, providing consistent evidence supporting the contextual distance curse (Hypothesis 1). The minor variations between formats do not alter the overall conclusion, though in-response rationales occasionally show slightly higher baseline performance, likely due to their alignment with default prompting structures.

Table 3: Performance comparison (accuracy in %) of in-response vs. in-prompt dummy rationales across varying rationale lengths on the MiniARC dataset, including the baseline of direct answering (contextual distance=0). Results are averaged across six LLMs.

| Approach | Direct (Dist.=0) | Count Down (100) | Count Down (400) | Shakespeare (1) | Shakespeare (4) |
|---|---|---|---|---|---|
| In Response | 16.89 | 14.77 (-2.12) | 13.65 (-3.24) | 14.77 (-2.12) | 12.98 (-3.91) |
| In Prompt | 16.89 | 13.98 (-2.91) | 12.42 (-4.47) | 13.98 (-2.91) | 13.42 (-3.47) |

Table 4: Performance comparison (accuracy in %) of in-response vs. in-prompt dummy rationales across varying rationale lengths on the COGS dataset, including the baseline of direct answering (contextual distance=0). Results are averaged across six LLMs.

| Approach | Direct (Dist.=0) | Count Down (100) | Count Down (400) | Shakespeare (1) | Shakespeare (4) |
|---|---|---|---|---|---|
| In Response | 16.55 | 16.24 (-0.31) | 14.90 (-1.65) | 15.77 (-0.78) | 13.76 (-2.79) |
| In Prompt | 16.55 | 14.77 (-1.78) | 12.42 (-4.13) | 13.98 (-2.57) | 13.42 (-3.13) |

### B.2 Rationale frontloading experiment

Another dimension of controlling the contextual distance effect involves retaining the semantic content of the CoT rationale while eliminating the contextual distance between demonstrations and the final answer (as in direct answering). To achieve this, we collect CoT rationales from the same models and prepend them to the in-context demonstrations of the question. Subsequently, we feed the combined input (question + inserted rationale) to the same LLMs for direct answering. However, we observed that CoT rationales sometimes already contain concluded answers. To include only the reasoning steps, we utilized GPT-4o-mini to process the rationales effectively with two-shot demonstrations, removing the final concluded answer while preserving the entire reasoning process.

The prompt instructions for rationale processing are provided below:

---

**Prompt Templates**

**Rationale Processing**

```
Demo1
User: <instruction> <CoT rationale with answer 1>
Assistant: <processed CoT rationale 1>

Demo2
User: <instruction> <CoT rationale with answer 2>
Assistant: <processed CoT rationale 2>

User: <instruction> <CoT rationale to process>
Assistant: __
```

---

**Rationale Frontloading**

```
<regular question instruction>

<processed CoT rationales>

<in-context demonstrations>

<regular answer instruction and test input (direct answering)>
```

---

### B.3    Pattern inference and execution experiment

The primary objective of this experiment is to disentangle the two stages of reasoning in CoT: pattern inference from in-context demonstrations and pattern execution on test inputs. The prompt instructions for both experiments are provided below:

---

**Prompt Templates**

**Pattern Inference**

```
<regular task description>
<in-context demonstrations>

Now, please perform reasoning to infer the underlying pattern (python function) mapping inputs and outputs.

Your output should strictly follow the json dict format below:
{
    "reasoning": "your reasoning steps",
    "pattern": "your pattern (function)"
}
```

---

**Pattern Execution**

```
<regular task description>
<test input>
<ground-truth pattern>

Now, please perform reasoning to apply the above ground-truth pattern to transform the input into output.

Your output should strictly follow the json dict format below:
{
    "reasoning": "your reasoning steps",
    "output": "your output"
}
```

---

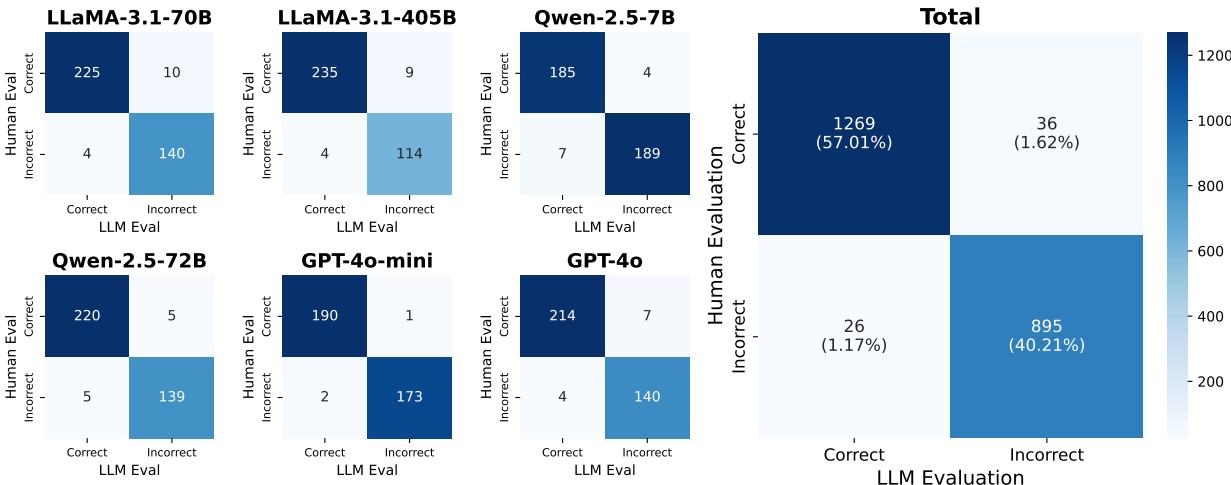

Figure 6: Alignment of human evaluation and LLM evaluation on inferred patterns.

The evaluation of inferred patterns in List Function and MiniSCAN is conducted using different approaches. For List Function, we directly execute Python functions generated by LLMs on all available input data and compare the program outputs with the corresponding ground-truth outputs. Over 95% of the LLM-generated programs successfully compile. For MiniSCAN, the generated rules are expressed as textual descriptions, which makes automated programmatic evaluation challenging. Therefore, we employ Qwen-2.5-72B to assess the correctness of the inferred rules. To evaluate the robustness of this LLM-based assessment, we also conducted human evaluation of rationales, demonstrating strong alignment between the results of LLM evaluation and human evaluation (97.22% total agreement, as shown in Figure 6). Consequently, our evaluation of inferred patterns is deemed reliable.

Below shows an example of the LLM pattern evaluation prompt for Qwen-2.5-72b:

---

**Prompt Templates**

**Pattern Evaluation**

```
You are tasked with judging a sequence-to-sequence problem.

A person is given a series of input and output sequences,
and aims to deduce the rules or word mappings that connect them.

In this scenario, each word in the input sequence can either:

1. Map directly to a word in the output sequence (word mapping).

2. Represent a rule for constructing the output sequence.

Possible Rules for Constructing the Output Sequence
Repeat the former part three times:
Example: If the input is "tmp thri" and "thri" represents this rule, the output should be "tmp tmp tmp."

Swap the former with the latter:
Example: If the input is "tmp1 sw tmp2" and "sw" represents this rule, the output should be "tmp2 tmp1."

Place the latter one between two instances of the former:
Example: If the input is "tmp1 pd tmp2" and "pd" represents this rule, the output should be "tmp1 tmp2 tmp1."

Your Task

You will be provided with the rules or word mappings that the person deduced.
Your objective is to evaluate whether the person correctly deduced these rules or mappings.
If a deduced rule only indicates that a word corresponds to a mapping or construction rule
without specifying what the rule is, it should be deemed incorrect.
```

---

### B.4 Decomposition of explicit and implicit reasoning

The inconsistency between overall CoT performance and the combined performance of explicit pattern inference and execution suggests that an implicit reasoning mechanism may also exist in the latent space of LLMs, despite the use of CoT. In Figure 5, we investigate, across all questions where CoT succeeds, how many questions the LLM fails to infer the correct pattern, while purely implicit reasoning (direct answering) succeeds, as well as the opposite case. These numbers provide intuitive yet practical estimates of the proportions of questions to which explicit and implicit reasoning contribute in CoT's success—proportions that are infeasible to evaluate at scale. For better understanding of this hybrid mechanism, we present a case study[4] for both datasets where CoT succeeds despite incorrect reasoning and inferred patterns. These serve as strong evidence of the contribution of latent reasoning in CoT to pattern-based ICL.

---

[4]For List Function, we required LLM to generate the python function between CoT reasoning steps and final answer output. In our regular evaluation, python functions are not explicitly required.

---

**Case Study (List Function)**

**List Function Prompt**

Below are several examples of input and output lists.
There exist a function that maps the input list to the output list.

Input1: [81, 18, 0, 41, 0, 89] Output1: [41, 0, 89]
Input2: [43, 47, 41, 83, 0, 22] Output2: [22]
Input3: [20, 89, 0, 53, 73] Output3: [53, 73]

Now, based on the new input list, please perform reasoning and provide the output list.

Input: [18, 91, 89, 21, 0, 53, 40, 93]

Generate the correct output. Please provide the python function that turns the input list into the output list.
Note that the python function should input a list and return a list.
```
{
    "reasoning": "your reasoning process",
    "function": "only the python function here",
    "output": your output list
}
```

---

**Ground Truth**

Python Function:

```
def func(x):
    # elements after the first 0
    return x[x.index(0)+1:]
```

Answer: [53, 40, 93]

---

**LLM Output (Llama-3.1-405B Instruct Turbo)**

Reasoning: (incorrect)
The function appears to be filtering out the numbers in the input list that are greater than the number that follows it. It seems to be iterating through the list from left to right, and whenever it encounters a number that is greater than the next number, it includes the next number in the output list. If the next number is 0, it skips the 0 and continues to the next non-zero number.

Function: (execution output: [89, 21, 53, 40])

```
def filter_list(input_list):
    output_list = []
    i = 0
    while i < len(input_list) - 1:
        if input_list[i] > input_list[i + 1]:
            if input_list[i + 1] != 0:
                output_list.append(input_list[i + 1])
            else:
                j = i + 2
                while j < len(input_list) and input_list[j] == 0:
                    j += 1
                    if j < len(input_list):\
                        output_list.append(input_list[j])
                        i = j - 1
        i += 1
    return output_list
```

Output: (correct)

[53, 40, 93]

---

## Case Study (MiniSCAN)

### MiniSCAN Prompt

Here is a task:Your task is to convert an input sequence into an output sequence based on specific rules. Each word
in the input sequence either corresponds to a direct transformation into a word in the output sequence, or a rule that defines
how the output sequence should be structured.

Below are some examples
Input1: schwuen Output1: bleently
Input2: synieusk Output2: flauw
Input3: syneaum Output3: hyaently
Input4: throds Output4: gloiark
Input5: synieusk glaently Output5: flauw flauw flauw
Input6: schwuen glaently Output6: bleently bleently bleently
Input7: synieusk zoully syneaum Output7: flauw hyaently flauw
Input8: syneaum zoully schwuen Output8: hyaently bleently hyaently
Input9: synieusk throks syneaum Output9: hyaently flauw
Input10: schwuen throks synieusk Output10: flauw bleently
Input11: synieusk glaently throks syneaum Output11: hyaently flauw flauw flauw
Input12: syneaum throks schwuen zoully synieusk Output12: bleently flauw bleently hyaently
Input13: synieusk throks syneaum glaently Output13: hyaently hyaently hyaently flauw
Input14: syneaum zoully schwuen throks synieusk Output14: flauw hyaently bleently hyaently
Now, based on the new input sequence, please provide the output sequence.
Input: throds zoully throds throks throds glaently
Your output should very strictly follow the json dict format below. All your answers must be within the curly braces:
{
"reasoning": "your reasoning steps",
"Output": "your output sequence"
}

---

### Ground Truth

Required Rules:

```
"throds": "gloiark",
"glaently": "Repeat the former three times",
"zoully": "Put the latter one between the two formers",
"throks": "Swap the former with the latter"
```

---

### LLM Output (Qwen-2.5-72B Instruct Turbo)

Reasoning: (rule incorrect, execution output: "gloiark gloiark flauw gloiark gloiark gloiark")

From the examples provided, we can deduce the following transformations: 'schwuen' -> 'bleently', 'synieusk' -> 'flauw',
'syneaum' -> 'hyaently', 'throds' -> 'gloiark', 'throks' -> 'flauw' when it follows 'synieusk' or 'schwuen', and 'zoully'
acts as a placeholder that does not change the word but affects the order or repetition. 'glaently' causes the previous word to
repeat three times. Applying these rules to the input 'throds zoully throds throks throds glaently':
1. 'throds' -> 'gloiark'
2. 'zoully' does not change the word but affects the order
3. 'throds' -> 'gloiark'
4. 'throks' -> 'flauw' (as it follows 'throds')
5. 'throds' -> 'gloiark'
6. 'glaently' causes the previous word 'gloiark' to repeat three times.

Output: (correct)

"gloiark gloiark gloiark gloiark gloiark gloiark"

### B.5 Pattern execution with in-context demonstrations

During pattern execution in the standard CoT setting, LLMs can not only deductively apply the inferred pattern to the test instance but also be implicitly influenced by the in-context demonstrations. However, these demonstrations are absent in the execution step of the "Inference & Execution" setting. To better demonstrate this impact, we conducted an additional experiment that incorporates in-context demonstrations into the pattern execution step. The experimental results from Table 5 and Table 6 show that performance significantly improves after incorporating in-context demonstrations into the deductive execution step, even though they are not directly required in the task setting. These findings serve as key evidence of CoT's reliance on implicit reasoning from demonstrations, in addition to explicit rationales.

Table 5: Experimental Results of "Inference & Execution" with in-context demonstrations provided in the pattern execution step, in dataset MiniSCAN.

| Model | CoT | Inference & Execution | Inference & Execution (execution with demos) |
|---|---|---|---|
| GPT-4o-mini | 4.10 | 1.35 | 2.81 (+1.46) |
| GPT-4o | 20.60 | 11.24 | 16.97 (+5.73) |
| Llama-3.1-70b | 19.80 | 8.89 | 14.49 (+5.60) |
| Llama-3.1-405b | 32.00 | 16.74 | 29.23 (+12.49) |
| Qwen-2.5-7b | 1.42 | 1.21 | 1.92 (+0.71) |
| Qwen-2.5-72b | 22.00 | 13.91 | 16.32 (+2.41) |
| Average | 16.65 | 8.89 | 13.62 (+4.73) |

Table 6: Experimental Results of "Inference & Execution" with in-context demonstrations provided in the pattern execution step, in dataset List Function.

| Model | CoT | Inference & Execution | Inference & Execution (execution with demos) |
|---|---|---|---|
| GPT-4o-mini | 36.88 | 31.00 | 37.81 (+6.81) |
| GPT-4o | 48.72 | 43.81 | 47.50 (+3.69) |
| Llama-3.1-70b | 37.92 | 31.35 | 37.92 (+6.57) |
| Llama-3.1-405b | 47.76 | 36.40 | 40.76 (+4.36) |
| Qwen-2.5-7b | 29.36 | 17.27 | 19.68 (+2.41) |
| Qwen-2.5-72b | 45.04 | 37.99 | 52.01 (+14.02) |
| Average | 40.95 | 32.97 | 39.28 (+6.31) |

## C  Dataset details

Among the nine datasets in our experiment, three were not originally designed for in-context learning in natural language processing. Here, we provide further details on the data processing:

- **COGS:** The original COGS dataset (Kim & Linzen, 2020) evaluates the compositional generalization of machine learning models through a task that introduces compositional distribution shifts in input-output mappings. In this study, we use the test dataset, sampling 10 entries as in-context demonstrations.

- **List Function:** The original work designed these functions to investigate the human-like learning abilities of cognitive systems (Rule, 2020). Subsequent studies have explored LLMs' capabilities in rule induction (pattern inference) (Qiu et al., 2024; Li et al., 2025) and in-context learning (output prediction) (Zheng et al., 2025a). In this work, we adopt the processed dataset from Li et al. (2025).

- **RAVEN:** The original RAVEN dataset (Zhang et al., 2019) assesses the analogical reasoning abilities of visual models using images of symbols. We adopt the abstracted lm-RAVEN dataset (Hu et al., 2023), which tokenizes image attributes into symbolic matrices.

# D   Prompt templates

In this section, we include our prompt templates as follows: prompt for dataset-specific instructions, prompt for reasoning frameworks, and prompt used in our tailored experiment.

## D.1   Prompt for dataset-specific instructions

---

**Prompt Templates**

**ARC-AGI / MiniARC / 1D-ARC / COGS**

Below are several examples of input and output grids/lists.
There exists an underlying pattern/function that maps the input grid/list to the output grid/list.

\<in-context demonstrations\>

Your task is to predict the output grid/list based on the new input grid/list:

\<test input\>

---

**SCAN**

Below are several examples that convert natural language commands into action sequences.

\<in-context demonstrations\>

Your task is to predict the output sequence based on the new input sequence:

\<test input\>

---

**MiniSCAN**

Here is a task:Your task is to convert an input sequence into an output sequence based on specific rules.
Each word in the input sequence either corresponds to a direct transformation into a word in the output sequence,
or a rule that defines how the output sequence should be structured.

\<in-context demonstrations\>

Your task is to predict the output sequence based on the new input sequence:

\<test input\>

---

**SALT**

Below are several examples that convert english sentence into an output sequence based on specific rules.
Each word in the input sequence either corresponds to a translated word in the output,
or indicates a syntactic rule (e.g., repeating or reordering semantic units) for forming the output sequence.

\<in-context demonstrations\>

Your task is to predict the output sequence based on the new english sentence:

\<test input\>

---

**List Function**

Below are several examples of input and output lists.
There exists an underlying python function that maps the input list to the output list.

\<in-context demonstrations\>

Your task is to predict the output list based on the new input list:

\<test input\>

---

**RAVEN**

Below are several rows of abstracted symbols. The symbols follow a certain rule or pattern.

\<in-context demonstrations\>

Your task is to predict the missing symbol based on the incomplete row:

\<test input\>

---

### D.2 Prompt for reasoning frameworks

For a fair comparison of reasoning frameworks against vanilla zero-shot CoT and direct answering, we adopt a one-off prompting approach rather than a complex agent framework. For Tree-of-Thought, we use the prompt proposed by Hulbert (2023). For ReAct, we employ the prompt from Qwen's implementation (Bai et al., 2023).

---

**Prompt Templates**

**Direct Answering**

```
<regular question instructions and data>

Please output your final answer in the following json dict format without any explanation:

{
    "answer": "your answer"
}
```

**Chain-of-Thought**

```
<regular question instructions and data>

Please first perform reasoning and then output your final answer in the following json dict format:

{
    "reasoning": "your reasoning process",
    "answer": "your answer"
}
```

**ReAct**

```
You should now solve the below question using the following pipeline:

Question: the input question you must answer
Thought: Think about what to do
Action: Your action process
Observation: the result of the action
(this Thought/Action/Observation can be repeated zero or more times)
Thought: I now know the final answer
Final Answer: the final answer to the original input question

<regular question instructions and data>

You should respond in the following json dict format:

{
    "process": "your full problem-solving process",
    "answer": "your final answer"
}
```

**Tree-of-Thought**

```
Imagine three different experts are answering this question.
All experts will write down 1 step of their thinking,
then share it with the group.
Then all experts will go on to the next step, etc.
If any expert realises they're wrong at any point then they leave.

<regular question instructions and data>

You should respond in the following json dict format:

{
    "discussion": "full discussion and reasoning process of experts",
    "answer": "final agreed answer"
}
```

---

# E   Full results

The detailed LLM performances on ICL benchmarks are presented in tables below:

- Table 7: ARC-AGI
- Table 8: MiniARC
- Table 9: 1D-ARC
- Table 10: SCAN
- Table 11: MiniSCAN
- Table 12: COGS
- Table 13: SALT
- Table 14: List Function
- Table 15: RAVEN

Table 7: Detailed LLM Performances on ARC-AGI.

| Model | Direct | CoT | | ReAct | | ToT | |
|---|---|---|---|---|---|---|---|
| | Acc (%) | Acc (%) | # tokens | Acc (%) | # tokens | Acc (%) | # tokens |
| *Open-source* | | | | | | | |
| Deepseek-V3 | 15.93 | 12.81 (-3.12) | 800.33 | 11.50 (-4.43) | 719.48 | 11.98 (-3.95) | 1055.27 |
| Llama3.1-8B | 3.95 | 2.75 (-1.20) | 1324.51 | 2.28 (-1.67) | 1735.61 | 2.16 (-1.79) | 2125.06 |
| Llama3.1-70B | 10.66 | 8.02 (-2.64) | 645.38 | 10.25 (-0.41) | 681.89 | 7.66 (-3.00) | 1477.78 |
| Llama3.1-405B | 16.45 | 10.42 (-6.03) | 699.35 | 11.86 (-4.59) | 1434.73 | 8.54 (-7.91) | 1147.96 |
| Qwen2.5-7B | 4.31 | 1.92 (-2.39) | 1681.24 | 0.96 (-3.35) | 1841.80 | 1.32 (-2.99) | 1988.00 |
| Qwen2.5-72B | 11.98 | 11.14 (-0.84) | 1021.80 | 1.80 (-10.18) | 1094.05 | 7.90 (-4.08) | 1430.62 |
| Mistral-7B | 0.36 | 0.48 (+0.12) | 672.39 | 0.96 (+0.60) | 758.50 | 0.48 (+0.12) | 902.04 |
| Mistral-Small 3 | 10.30 | 5.99 (-4.31) | 1768.28 | 0.72 (-9.58) | 409.83 | 5.15 (-5.15) | 1619.02 |
| *Proprietary* | | | | | | | |
| Gemini-1.5-flash | 11.26 | 7.90 (-3.36) | 727.77 | 12.33 (+1.07) | 872.89 | 13.11 (+1.85) | 930.87 |
| Gemini-1.5-pro | 17.25 | 13.41 (-3.84) | 787.59 | 4.08 (-13.17) | 1080.24 | 15.15 (-2.10) | 840.94 |
| Gemini-2.0-flash | 14.25 | 10.06 (-4.19) | 867.74 | 11.67 (-2.58) | 1005.52 | 9.34 (-4.91) | 3645.00 |
| GPT-3.5-turbo | 4.09 | 4.55 (+0.46) | 459.40 | 3.29 (-0.80) | 213.74 | 2.44 (-1.65) | 255.09 |
| GPT-4o-mini | 5.51 | 5.15 (-0.36) | 632.05 | 4.01 (-1.50) | 754.46 | 3.71 (-1.80) | 840.72 |
| GPT-4o | 13.77 | 10.42 (-3.35) | 708.77 | 11.55 (-2.22) | 777.92 | 8.98 (-4.79) | 1019.13 |
| **Average** | 10.01 | 7.50 (-2.51) | 914.04 | 6.23 (-3.78) | 955.76 | 6.99 (-3.02) | 1376.96 |

Table 8: Detailed LLM Performances on MiniARC.

| Model | Direct | CoT | | ReAct | | ToT | |
|---|---|---|---|---|---|---|---|
| | Acc (%) | Acc (%) | # tokens | Acc (%) | # tokens | Acc (%) | # tokens |
| *Open-source* | | | | | | | |
| Deepseek-V3 | 27.52 | 15.44 (-12.08) | 710.46 | 15.44 (-12.08) | 789.48 | 14.77 (-12.75) | 395.23 |
| Gemma2-9B | 11.41 | 3.36 (-8.05) | 194.93 | 1.34 (-10.07) | 435.62 | 2.68 (-8.73) | 138.50 |
| Gemma2-27B | 14.09 | 11.41 (-2.68) | 142.81 | 3.36 (-10.73) | 332.99 | 6.71 (-7.38) | 114.46 |
| Llama3.1-8B | 10.74 | 6.71 (-4.03) | 566.13 | 4.70 (-6.04) | 1537.84 | 3.36 (-7.38) | 323.08 |
| Llama3.1-70B | 21.48 | 14.09 (-7.39) | 173.00 | 13.42 (-8.06) | 816.99 | 8.72 (-12.76) | 126.50 |
| Llama3.1-405B | 26.71 | 16.11 (-10.60) | 522.15 | 15.44 (-11.27) | 645.18 | 14.09 (-12.62) | 301.58 |
| Qwen2.5-7B | 10.07 | 2.01 (-8.06) | 587.68 | 0.00 (-10.07) | 811.81 | 4.03 (-6.04) | 335.85 |
| Qwen2.5-72B | 23.49 | 7.38 (-16.11) | 592.43 | 0.67 (-22.82) | 881.40 | 10.07 (-13.42) | 336.21 |
| Mistral-7B | 2.68 | 0.67 (-2.01) | 173.39 | 2.68 (0.00) | 356.87 | 0.67 (-2.01) | 108.88 |
| Mistral-Small 3 | 16.11 | 6.71 (-9.40) | 805.96 | 2.01 (-14.10) | 750.36 | 9.40 (-6.71) | 445.04 |
| *Proprietary* | | | | | | | |
| Gemini-1.5-flash | 16.11 | 14.09 (-2.02) | 278.79 | 10.74 (-5.37) | 592.56 | 10.07 (-6.04) | 181.90 |
| Gemini-1.5-pro | 24.83 | 21.48 (-3.35) | 626.63 | 15.44 (-9.39) | 460.40 | 16.78 (-8.05) | 205.33 |
| Gemini-2.0-flash | 25.50 | 14.09 (-11.41) | 626.63 | 18.12 (-7.38) | 1048.62 | 11.41 (-14.09) | 246.71 |
| GPT-3.5-turbo | 7.38 | 5.37 (-2.01) | 153.30 | 4.70 (-2.68) | 193.79 | 6.04 (-1.34) | 116.19 |
| GPT-4o-mini | 13.42 | 10.74 (-2.68) | 252.89 | 11.41 (-2.01) | 407.54 | 8.05 (-5.37) | 165.94 |
| GPT-4o | 22.15 | 16.11 (-6.04) | 308.75 | 19.61 (-2.54) | 552.40 | 14.77 (-7.38) | 194.06 |
| **Average** | 17.11 | 10.36 (-6.75) | 419.75 | 8.69 (-8.42) | 663.37 | 8.85 (-8.26) | 233.47 |

Table 9: Detailed LLM Performances on 1D-ARC.

| Model | Direct | CoT | | ReAct | | ToT | |
|---|---|---|---|---|---|---|---|
| | Acc (%) | Acc (%) | # tokens | Acc (%) | # tokens | Acc (%) | # tokens |
| *Open-source* | | | | | | | |
| Deepseek-V3 | 69.70 | 66.26 (-3.44) | 723.92 | 66.93 (-2.77) | 775.21 | 55.38 (-14.32) | 733.13 |
| Gemma2-9B | 28.30 | 20.20 (-8.10) | 171.04 | 11.54 (-16.76) | 239.97 | 11.10 (-17.20) | 357.99 |
| Gemma2-27B | 41.62 | 31.08 (-10.54) | 141.58 | 22.42 (-19.20) | 212.37 | 23.75 (-17.87) | 287.10 |
| Llama3.1-8B | 23.20 | 13.21 (-9.99) | 484.33 | 12.32 (-10.88) | 873.66 | 10.65 (-12.55) | 1349.24 |
| Llama3.1-70B | 53.89 | 46.00 (-7.89) | 165.14 | 40.51 (-13.38) | 301.79 | 33.96 (-19.93) | 697.90 |
| Llama3.1-405B | 60.60 | 58.49 (-2.11) | 434.75 | 55.83 (-4.77) | 665.89 | 44.28 (-16.32) | 677.26 |
| Qwen2.5-7B | 25.86 | 13.67 (-12.19) | 445.33 | 11.76 (-14.10) | 506.17 | 3.88 (-21.98) | 705.43 |
| Qwen2.5-72B | 51.67 | 48.22 (-3.45) | 294.11 | 13.32 (-38.35) | 353.90 | 36.40 (-15.27) | 914.23 |
| Mistral-7B | 0.00 | 1.00 (+1.00) | 193.50 | 1.22 (+1.22) | 334.70 | 1.11 (+1.11) | 341.51 |
| Mistral-Small 3 | 47.50 | 32.74 (-14.76) | 870.63 | 0.00 (-47.50) | 480.67 | 29.74 (-17.76) | 694.24 |
| *Proprietary* | | | | | | | |
| Gemini-1.5-flash | 53.27 | 39.84 (-13.43) | 231.44 | 40.51 (-12.76) | 466.01 | 34.07 (-19.20) | 538.92 |
| Gemini-1.5-pro | 67.04 | 58.71 (-8.33) | 269.24 | 56.27 (-10.77) | 420.13 | 48.95 (-18.09) | 388.57 |
| Gemini-2.0-flash | 60.38 | 50.94 (-9.44) | 644.63 | 48.83 (-11.55) | 452.14 | 45.51 (-14.87) | 748.79 |
| GPT-3.5-turbo | 8.66 | 14.43 (+5.77) | 174.62 | 15.32 (+6.66) | 142.77 | 11.43 (+2.77) | 183.69 |
| GPT-4o-mini | 26.53 | 19.64 (-6.89) | 227.64 | 17.76 (-8.77) | 379.46 | 17.43 (-9.10) | 401.00 |
| GPT-4o | 42.51 | 44.40 (+1.89) | 281.26 | 41.62 (-0.89) | 370.70 | 38.40 (-4.11) | 496.21 |
| **Average** | 41.30 | 34.93 (-6.37) | 359.57 | 28.51 (-12.79) | 435.97 | 27.88 (-13.42) | 594.70 |

Table 10: Detailed LLM Performances on SCAN.

| Model | Direct | CoT | | ReAct | | ToT | |
|---|---|---|---|---|---|---|---|
| | Acc (%) | Acc (%) | # tokens | Acc (%) | # tokens | Acc (%) | # tokens |
| *Open-source* | | | | | | | |
| Deepseek-V3 | 91.85 | 81.70 (-10.15) | 168.76 | 81.90 (-9.95) | 248.69 | 78.00 (-13.85) | 225.57 |
| Gemma2-9B | 32.60 | 38.60 (+6.00) | 85.34 | 36.30 (+3.70) | 169.07 | 28.17 (-4.43) | 352.10 |
| Gemma2-27B | 60.86 | 53.70 (-7.16) | 102.49 | 55.30 (-5.56) | 242.08 | 46.46 (-14.40) | 430.27 |
| Llama3.1-8B | 24.92 | 23.79 (-1.13) | 142.52 | 14.19 (-10.73) | 659.83 | 14.41 (-10.51) | 1003.49 |
| Llama3.1-70B | 68.38 | 60.26 (-8.12) | 233.15 | 55.40 (-12.98) | 525.03 | 53.70 (-14.68) | 1080.48 |
| Llama3.1-405B | 84.42 | 81.70 (-2.72) | 154.70 | 86.00 (+1.58) | 336.10 | 79.20 (-5.22) | 646.67 |
| Qwen2.5-7B | 41.32 | 33.53 (-7.79) | 115.28 | 31.89 (-9.43) | 174.10 | 31.16 (-10.16) | 300.71 |
| Qwen2.5-72B | 88.05 | 88.55 (+0.50) | 99.52 | 89.48 (+1.43) | 204.55 | 87.00 (-1.05) | 180.86 |
| Mistral-7B | 21.21 | 17.07 (-4.14) | 122.67 | 20.90 (-0.31) | 163.15 | 11.01 (-10.20) | 270.43 |
| Mistral-Small 3 | 70.90 | 67.75 (-3.15) | 156.90 | 52.10 (-18.80) | 335.95 | 43.89 (-27.01) | 500.01 |
| *Proprietary* | | | | | | | |
| Gemini-1.5-flash | 69.30 | 59.80 (-9.50) | 113.15 | 56.60 (-12.70) | 236.42 | 46.50 (-22.80) | 342.88 |
| Gemini-1.5-pro | 93.10 | 87.20 (-5.90) | 155.83 | 87.00 (-6.10) | 225.16 | 71.00 (-22.10) | 228.23 |
| Gemini-2.0-flash | 86.30 | 86.20 (-0.10) | 161.69 | 81.30 (-5.00) | 355.34 | 76.70 (-9.60) | 951.81 |
| GPT-3.5-turbo | 37.60 | 41.00 (+3.40) | 77.67 | 28.50 (-9.10) | 73.53 | 22.80 (-14.80) | 198.80 |
| GPT-4o-mini | 51.00 | 57.40 (+6.40) | 115.21 | 57.20 (+6.20) | 162.19 | 48.20 (-2.80) | 263.84 |
| GPT-4o | 82.90 | 82.40 (-0.50) | 147.24 | 83.60 (+0.70) | 211.43 | 82.70 (-0.20) | 310.11 |
| **Average** | 62.79 | 60.04 (-2.75) | 134.51 | 57.35 (-5.44) | 270.16 | 51.31 (-11.48) | 455.39 |

Table 11: Detailed LLM Performances on MiniSCAN.

| Model | Direct | CoT | | ReAct | | ToT | |
|---|---|---|---|---|---|---|---|
| | Acc (%) | Acc (%) | # tokens | Acc (%) | # tokens | Acc (%) | # tokens |
| *Open-source* | | | | | | | |
| Deepseek-V3 | 16.50 | 16.30 (-0.20) | 252.12 | 18.80 (+2.30) | 285.60 | 20.80 (+4.30) | 258.93 |
| Gemma2-9B | 9.50 | 3.01 (-6.49) | 149.56 | 1.00 (-8.50) | 190.38 | 0.80 (-8.70) | 403.07 |
| Gemma2-27B | 16.40 | 13.50 (-2.90) | 107.12 | 8.10 (-8.30) | 239.84 | 7.90 (-8.50) | 485.17 |
| Llama3.1-8B | 1.10 | 2.30 (+1.20) | 289.07 | 0.47 (-0.63) | 737.72 | 2.60 (+1.50) | 1226.43 |
| Llama3.1-70B | 27.00 | 19.80 (-7.20) | 365.74 | 17.71 (-9.29) | 345.85 | 16.70 (-10.30) | 1067.96 |
| Llama3.1-405B | 33.30 | 32.00 (-1.30) | 383.64 | 35.00 (+1.70) | 385.51 | 24.60 (-8.70) | 741.70 |
| Qwen2.5-7B | 2.80 | 1.42 (-1.38) | 146.22 | 1.34 (-1.46) | 182.02 | 3.77 (+0.97) | 881.11 |
| Qwen2.5-72B | 20.00 | 22.00 (+2.00) | 204.60 | 20.70 (+0.70) | 261.85 | 19.50 (-0.50) | 316.75 |
| Mistral-7B | 0.20 | 1.00 (+0.80) | 182.66 | 1.13 (+0.93) | 198.82 | 0.58 (+0.38) | 348.98 |
| Mistral-Small 3 | 29.63 | 28.30 (-1.33) | 385.82 | 23.50 (-6.13) | 297.09 | 30.59 (+0.96) | 575.90 |
| *Proprietary* | | | | | | | |
| Gemini-1.5-flash | 35.40 | 32.10 (-3.30) | 233.42 | 28.00 (-7.40) | 523.10 | 23.00 (-12.40) | 573.68 |
| Gemini-1.5-pro | 46.80 | 45.75 (-1.05) | 299.28 | 42.60 (-4.20) | 523.00 | 41.00 (-5.80) | 258.07 |
| Gemini-2.0-flash | 47.30 | 31.30 (-16.00) | 329.51 | 31.20 (-16.10) | 523.10 | 32.10 (-15.20) | 745.29 |
| GPT-3.5-turbo | 6.10 | 1.40 (-4.70) | 98.67 | 0.30 (-5.80) | 103.16 | 1.50 (-4.60) | 261.70 |
| GPT-4o-mini | 6.30 | 4.10 (-2.20) | 160.05 | 2.80 (-3.50) | 208.32 | 1.60 (-4.70) | 329.40 |
| GPT-4o | 35.80 | 20.60 (-15.20) | 252.39 | 18.90 (-16.90) | 276.85 | 19.60 (-16.20) | 399.70 |
| **Average** | 20.88 | 17.18 (-3.70) | 239.99 | 15.72 (-5.16) | 330.14 | 15.42 (-5.46) | 554.62 |

Table 12: Detailed LLM Performances on COGS.

| Model | Direct | CoT | | ReAct | | ToT | |
|---|---|---|---|---|---|---|---|
| | Acc (%) | Acc (%) | # tokens | Acc (%) | # tokens | Acc (%) | # tokens |
| *Open-source* | | | | | | | |
| Deepseek-V3 | 30.80 | 14.00 (-16.80) | 405.03 | 27.80 (-3.00) | 375.88 | 15.10 (-15.70) | 729.94 |
| Gemma2-9B | 9.10 | 8.50 (-0.60) | 309.71 | 6.70 (-2.40) | 223.18 | 1.30 (-7.80) | 328.61 |
| Gemma2-27B | 21.30 | 14.10 (-7.20) | 132.83 | 11.40 (-9.90) | 243.44 | 7.90 (-13.40) | 267.56 |
| Llama3.1-8B | 8.10 | 3.60 (-4.50) | 415.50 | 2.80 (-5.30) | 628.31 | 2.40 (-5.70) | 1072.55 |
| Llama3.1-70B | 20.80 | 17.40 (-3.40) | 207.46 | 12.20 (-8.60) | 348.43 | 4.70 (-16.10) | 669.70 |
| Llama3.1-405B | 24.40 | 13.80 (-10.60) | 304.01 | 8.20 (-16.20) | 395.69 | 3.20 (-21.20) | 693.21 |
| Qwen2.5-7B | 11.50 | 9.90 (-1.60) | 241.29 | 7.10 (-4.40) | 310.74 | 5.00 (-6.50) | 511.24 |
| Qwen2.5-72B | 20.40 | 19.20 (-1.20) | 196.61 | 12.40 (-8.00) | 86.36 | 17.50 (-2.90) | 593.44 |
| Mistral-7B | 7.60 | 5.20 (-2.40) | 152.42 | 2.70 (-4.90) | 139.00 | 3.10 (-4.50) | 163.96 |
| Mistral-Small 3 | 15.40 | 12.70 (-2.70) | 369.97 | 12.90 (-2.50) | 192.04 | 8.30 (-7.10) | 595.81 |
| *Proprietary* | | | | | | | |
| Gemini-1.5-flash | 23.70 | 19.20 (-4.50) | 181.79 | 11.50 (-12.20) | 293.78 | 4.95 (-18.75) | 459.69 |
| Gemini-1.5-pro | 36.00 | 28.10 (-7.90) | 226.60 | 25.29 (-10.71) | 344.10 | 22.50 (-13.50) | 463.41 |
| Gemini-2.0-flash | 31.60 | 28.70 (-2.90) | 231.77 | 25.20 (-6.40) | 216.32 | 14.81 (-16.79) | 425.68 |
| GPT-3.5-turbo | 11.60 | 8.40 (-3.20) | 113.25 | 8.50 (-3.10) | 116.25 | 7.89 (-3.71) | 138.12 |
| GPT-4o-mini | 18.10 | 13.50 (-4.60) | 171.29 | 11.90 (-6.20) | 233.30 | 10.10 (-8.00) | 247.97 |
| GPT-4o | 25.30 | 21.70 (-3.60) | 246.19 | 21.30 (-4.00) | 208.00 | 19.16 (-6.14) | 397.85 |
| **Average** | 19.73 | 14.88 (-4.85) | 244.11 | 12.99 (-6.74) | 272.18 | 9.24 (-10.49) | 484.92 |

Table 13: Detailed LLM Performances on SALT.

| Model | Direct | CoT | | ReAct | | ToT | |
|---|---|---|---|---|---|---|---|
| | Acc (%) | Acc (%) | # tokens | Acc (%) | # tokens | Acc (%) | # tokens |
| *Open-source* | | | | | | | |
| Deepseek-V3 | 52.83 | 45.83 (-7.00) | 161.16 | 36.08 (-16.75) | 419.34 | 41.92 (-10.91) | 549.59 |
| Gemma2-9B | 13.83 | 19.17 (+5.34) | 139.96 | 17.08 (+3.25) | 270.40 | 15.33 (+1.50) | 294.55 |
| Gemma2-27B | 39.33 | 34.33 (-5.00) | 100.25 | 26.67 (-12.66) | 222.94 | 11.17 (-28.16) | 323.54 |
| Llama3.1-8B | 25.25 | 17.92 (-7.33) | 253.14 | 15.17 (-10.08) | 505.72 | 15.00 (-10.25) | 1082.77 |
| Llama3.1-70B | 48.33 | 44.92 (-3.41) | 204.69 | 41.42 (-6.91) | 339.29 | 37.67 (-10.66) | 602.01 |
| Llama3.1-405B | 57.92 | 59.67 (+1.75) | 430.02 | 54.17 (-3.75) | 435.96 | 33.50 (-24.42) | 616.12 |
| Qwen2.5-7B | 15.58 | 13.33 (-2.25) | 116.37 | 11.00 (-4.58) | 165.47 | 10.83 (-4.75) | 318.66 |
| Qwen2.5-72B | 35.25 | 39.42 (+4.17) | 137.75 | 38.33 (+3.08) | 240.15 | 43.50 (+8.25) | 324.49 |
| Mistral-7B | 14.92 | 11.75 (-3.17) | 112.82 | 11.17 (-3.75) | 148.71 | 5.42 (-9.50) | 357.95 |
| Mistral-Small 3 | 44.67 | 36.25 (-8.42) | 167.56 | 33.17 (-11.50) | 274.82 | 28.67 (-16.00) | 584.81 |
| *Proprietary* | | | | | | | |
| Gemini-1.5-flash | 39.50 | 38.08 (-1.42) | 201.34 | 41.58 (+2.08) | 456.13 | 34.42 (-5.08) | 542.69 |
| Gemini-1.5-pro | 53.50 | 45.92 (-7.58) | 213.63 | 43.83 (-9.67) | 476.21 | 42.92 (-10.58) | 387.65 |
| Gemini-2.0-flash | 53.67 | 48.92 (-4.75) | 143.62 | 51.08 (-2.59) | 403.63 | 47.50 (-6.17) | 762.87 |
| GPT-3.5-turbo | 26.50 | 20.33 (-6.17) | 107.55 | 15.50 (-11.00) | 149.05 | 15.00 (-11.50) | 154.97 |
| GPT-4o-mini | 31.83 | 24.17 (-7.66) | 151.52 | 18.08 (-13.75) | 245.74 | 11.67 (-20.16) | 540.46 |
| GPT-4o | 50.67 | 46.41 (-4.26) | 174.41 | 42.67 (-8.00) | 309.02 | 41.42 (-9.25) | 440.49 |
| **Average** | 37.72 | 34.15 (-3.57) | 175.99 | 31.06 (-6.66) | 316.41 | 27.25 (-10.47) | 492.73 |

Table 14: Detailed LLM Performances on List Function.

| Model | Direct | CoT | | ReAct | | ToT | |
|---|---|---|---|---|---|---|---|
| | Acc (%) | Acc (%) | # tokens | Acc (%) | # tokens | Acc (%) | # tokens |
| *Open-source* | | | | | | | |
| Deepseek-V3 | 54.88 | 58.32 (+3.44) | 487.50 | 34.56 (-20.32) | 457.90 | 47.20 (-7.68) | 476.20 |
| Gemma2-9B | 32.80 | 25.20 (-7.60) | 146.12 | 22.24 (-10.56) | 193.28 | 21.04 (-11.76) | 370.36 |
| Gemma2-27B | 43.60 | 35.44 (-8.16) | 98.08 | 34.88 (-8.72) | 186.05 | 36.40 (-7.20) | 243.89 |
| Llama3.1-8B | 35.20 | 24.48 (-10.72) | 435.90 | 18.48 (-16.72) | 621.80 | 15.12 (-20.08) | 978.29 |
| Llama3.1-70B | 41.28 | 37.92 (-3.36) | 171.67 | 38.16 (-3.12) | 312.06 | 38.40 (-2.88) | 578.48 |
| Llama3.1-405B | 50.40 | 47.76 (-2.64) | 425.44 | 47.68 (-2.72) | 410.83 | 19.68 (-30.72) | 545.26 |
| Qwen2.5-7B | 37.60 | 29.36 (-8.24) | 382.33 | 21.36 (-16.24) | 144.06 | 24.64 (-12.96) | 348.04 |
| Qwen2.5-72B | 49.28 | 45.04 (-4.24) | 427.86 | 42.96 (-6.32) | 266.22 | 41.92 (-7.36) | 380.14 |
| Mistral-7B | 28.00 | 8.96 (-19.04) | 161.29 | 4.96 (-23.04) | 129.42 | 2.00 (-26.00) | 392.23 |
| Mistral-Small 3 | 41.76 | 38.32 (-3.44) | 510.16 | 36.00 (-5.76) | 224.67 | 32.32 (-9.44) | 524.35 |
| *Proprietary* | | | | | | | |
| Gemini-1.5-flash | 46.96 | 42.00 (-4.96) | 369.28 | 42.72 (-4.24) | 485.17 | 39.44 (-7.52) | 567.06 |
| Gemini-1.5-pro | 53.28 | 52.00 (-1.28) | 341.34 | 53.60 (+0.32) | 447.85 | 46.32 (-6.96) | 456.78 |
| Gemini-2.0-flash | 53.76 | 51.12 (-2.64) | 332.87 | 53.04 (-0.72) | 451.64 | 40.56 (-13.20) | 895.93 |
| GPT-3.5-turbo | 42.16 | 31.12 (-11.04) | 131.08 | 25.92 (-16.24) | 99.82 | 24.48 (-17.68) | 154.43 |
| GPT-4o-mini | 44.88 | 36.88 (-8.00) | 188.62 | 35.04 (-9.84) | 228.55 | 34.56 (-10.32) | 331.80 |
| GPT-4o | 53.04 | 48.72 (-4.32) | 211.48 | 45.84 (-7.20) | 311.21 | 35.92 (-17.12) | 444.89 |
| **Average** | 44.58 | 38.80 (-5.78) | 305.49 | 35.25 (-9.33) | 310.73 | 31.63 (-12.95) | 486.49 |

Table 15: Detailed LLM Performances on RAVEN.

| Model | Direct | CoT | | ReAct | | ToT | |
|---|---|---|---|---|---|---|---|
| | Acc (%) | Acc (%) | # tokens | Acc (%) | # tokens | Acc (%) | # tokens |
| *Open-source* | | | | | | | |
| Deepseek-V3 | 21.05 | 8.98 (-12.07) | 397.14 | 6.27 (-14.78) | 413.69 | 2.14 (-18.91) | 780.90 |
| Gemma2-9B | 13.74 | 1.99 (-11.75) | 252.07 | 0.87 (-12.87) | 298.49 | 0.08 (-13.66) | 435.18 |
| Gemma2-27B | 16.60 | 1.11 (-15.49) | 196.51 | 1.91 (-14.69) | 287.23 | 0.24 (-16.36) | 349.96 |
| Llama3.1-8B | 7.07 | 0.87 (-6.20) | 456.49 | 1.43 (-5.64) | 757.91 | 0.32 (-6.75) | 943.80 |
| Llama3.1-70B | 17.08 | 9.93 (-7.15) | 325.79 | 5.24 (-11.84) | 576.85 | 1.19 (-15.89) | 1022.81 |
| Llama3.1-405B | 25.34 | 16.92 (-8.42) | 355.79 | 11.68 (-13.66) | 543.75 | 3.97 (-21.37) | 843.70 |
| Qwen2.5-7B | 11.12 | 0.56 (-10.56) | 673.17 | 1.59 (-9.53) | 545.11 | 0.24 (-10.88) | 964.45 |
| Qwen2.5-72B | 23.67 | 8.18 (-15.49) | 476.07 | 6.35 (-17.32) | 588.40 | 4.92 (-18.75) | 608.39 |
| Mistral-7B | 1.51 | 0.00 (-1.51) | 296.05 | 0.08 (-1.43) | 326.30 | 0.16 (-1.35) | 512.38 |
| Mistral-Small 3 | 16.44 | 10.41 (-6.03) | 828.07 | 5.80 (-10.64) | 475.78 | 2.14 (-14.30) | 841.24 |
| *Proprietary* | | | | | | | |
| Gemini-1.5-flash | 19.06 | 5.88 (-13.18) | 478.57 | 3.02 (-16.04) | 644.90 | 1.27 (-17.79) | 666.44 |
| Gemini-1.5-pro | 24.31 | 11.12 (-13.19) | 545.27 | 12.87 (-11.44) | 728.48 | 7.94 (-16.37) | 564.41 |
| Gemini-2.0-flash | 23.35 | 19.78 (-3.57) | 719.08 | 20.02 (-3.33) | 1017.31 | 13.66 (-9.69) | 1634.67 |
| GPT-3.5-turbo | 12.79 | 5.80 (-6.99) | 206.14 | 3.57 (-9.22) | 131.61 | 3.73 (-9.06) | 235.38 |
| GPT-4o-mini | 15.65 | 6.12 (-9.53) | 322.94 | 3.18 (-12.47) | 337.58 | 0.95 (-14.70) | 598.97 |
| GPT-4o | 22.24 | 10.33 (-11.91) | 427.07 | 8.90 (-13.34) | 469.12 | 6.43 (-15.81) | 718.92 |
| **Average** | 17.25 | 7.37 (-9.88) | 434.75 | 5.87 (-11.38) | 533.09 | 2.84 (-14.41) | 737.64 |

