# OpenReview forum: "The Curse of CoT: On the Limitations of Chain-of-Thought in In-Context Learning"
_TMLR — Accepted by TMLR_

### Review · Reviewer_MEMh · 2025-07-28

**Summary Of Contributions:**

A key strategy for improving the performance of LLMs on many tasks is "Chain of Thought" (CoT), where the model is prompted to generate tokens corresponding to some reasoning process before its final answer. CoT is generally viewed as an unalloyed good for performance, with the only downside being that it is more costly in terms of tokens. This paper shows interesting empirical evidence that in at least one case (in-context learning), it can actually harm performance, which is surprising.

In-context learning is the case where a few example inputs and outputs are put into the LLM's context, before generating an output for a new unlabeled example. In other words, without touching model weights, the LLM itself can function as a supervised learning algorithm. Observing this phenomenon in GPT models was an early driver of excitement about LLMs.

What the authors show is that, compared to direct output, adding various CoT methods to an extremely wide range of models doing in-context learning on a range of tasks hurts performance.

**Additional Comments:**

I've worked with LLMs in my research, including chain-of-thought prompting them to do various things and trying to get them to do in-context learning. So I'm not unfamiliar with the ideas here. However, empirical eval of LLMs is a deep and tricky area where I don't have special expertise. Therefore, if e.g. the chosen in-context datasets were somehow cherry-picked, if some essential datasets were left out, or if some special choices in prompting or hyperparameters explain these results, I wouldn't be able to catch this.

**Audience:**

Yes

**Audience Explanation:**

A lot of people are interested in LLMs as few-shot learners, even if just as end users. I think the results in this paper would be of wide interest.

**Broader Impact Concerns:**

No broader impact concerns in my view.

**Claims And Evidence:**

Yes

**Claims Explanation:**

Yes, the authors consider a wide range of models which are not that old, and they consider wide range of in-context learning tasks. They propose multiple hypotheses and rule out some. They also present convincing, though not totally certain, evidence explaining why CoT might interfere with in-context learning. The decline in performance from CoT is remarkably consistent across many settings. Figures 1 and 2 are very striking. The results are quite convincing, at least to me.

**Requested Changes:**

This is not strictly a request, just a remark: when I see the word "duality", it naturally makes me expect to see a rigorous mathematical concept. Personally, I would consider using a different name for the relationship between implicit and explicit reasoning.

---

> ### Author Response · Authors · 2025-09-27
>
> Thank you for your thoughtful review and positive feedback. We appreciate you highlighting the surprising nature of our findings and the convincing evidence in our hypothesis testing experiments.
>
> - **Regarding "duality"**: This is a great point. We agree the term can be misleading and will change the wording to avoid confusion.
> - **Regarding experimental scope**: We share your perspective on the challenges of LLM evaluation. This is precisely why we conducted such a large-scale study to ensure the robustness of our findings. From our experience, such performance decline is a consistent phenomenon, not substantially affected by prompt-level adjustments or temperature settings.
>
> Thank you again for your valuable feedback.

---

### Review · Reviewer_2yea · 2025-08-03

**Summary Of Contributions:**

The paper „The Curse of CoT: On the Limitations of Chain-of-Thought in In-Context Learning” describes an empirical study on problems CoT has with few-shot learning. The core of the paper is built on the foundation that CoT solutions underperform direct answering for few-shot tasks for multiple benchmarks. The authors then hypothesize about the reasons, i.e. that that the distance between few-shot examples and response created through the CoT is responsible, that LLMs fail to identify the pattern within the few-shot examples as part of the CoT, and that even if they can identify this pattern, they fail at applying the pattern. Based on the evaluation of these hypotheses, the authors finally look at how often LLMs are better with a direct response (implicit reasoning) in comparison to how often CoT actually helps (explicit reasoning) and find that the direct relationship is the strong, more important signal in comparison to the CoT.

**Audience:**

Yes

**Audience Explanation:**

CoT is a standard technique and knowledge about the limitations is important to guide expectations.

**Claims And Evidence:**

No

**Claims Explanation:**

I agree with the premise of the paper, i.e., that CoT is no guarantee to achieve better results in all settings. I also believe the authors provide strong evidence for this. I also like the general approach to then look at different hypotheses and test them to understand what might cause the, for many unexpected, behavior that CoT decreases performance. I think the analysis for H4 is very well done and the core contribution of this paper.

Major issues:

1) I do not think that the experiment conducted to assess H1 is suitable. The experiment only evaluates if the LLM performance degenerates, if the LLM is forced to generate noise before answering. Frankly, I am also not sure how to setup a better experiment. But simply stating: “if LLMs generate noise in between and the performance suffers, this explains that the performance also suffers when we ask LLMs to generate contextual information before coming to a result” is too much of a stretch for me. Unless the authors can come up with a better way to measure something related to H1, I suggest dropping this hypothesis.

2) Hypothesis 2 and Hypothesis 3 are independent from each other. Consequently, a statement like “Hypothesis 2 is validated over Hypothesis 3” does not make sense. Both should be independently accepted or rejected.

3) The results presented in Section 6 for LRMs are weak due to the unfitting study design. To really quantify the difference between a normal LLM and a LRM, one would need to have pairs of models, such that you have a LLM and a LRM from the same model family with roughly the same size. Otherwise, any comparison is very noisy because other factors (e.g., different pre-training, different model size, etc.) can explain the differences. The current setting is hardly fair. For example, Qwen2.5-72B is used as normal LLM, but the smaller QwQ-32B is used as reasoning counterpart.

Suggestions for additional improvements:

1) The paper anthropomorphizes the LLMs quite a lot. I suggest avoiding concepts like cognitive load for the stochastic reasoning modeling that LLMs implement or call what LLMs do “verbalization”.

**Requested Changes:**

See major issues above.

---

> ### Author Response · Authors · 2025-09-27
>
> Thanks for the insightful and constructive feedback. We are encouraged that the reviewer finds our premise strong and our analysis convincing, and we appreciate the opportunity to clarify our work based on the points raised.
>
> ---
>
> ### 1. On the Validity of the H1 Experiment (Contextual Distance):
>
> We thank the reviewer for pushing us to justify this methodology. Our use of dummy tokens to isolate the effects of contextual distance from semantic content is an established method in research on long-context utilization in LLMs [1, 2, 3]. To ensure robustness, we also ran extensive controls:
> *   We varied the dummy token format (textual via "shakespeare" vs. numerical via "count-down").
> *   We varied the dummy token position (in-prompt vs. in-response, detailed in Appendix B.1.2).
> *   We ran a "rationale frontloading" experiment that modifies contextual distance *without* involving any dummy tokens. **Even in this setup, which contains no "noise" information, the performance degradation from increased distance persists.**
>
> Across all controlled conditions, the results consistently supported H1. We are confident this conclusion is well-supported.
>
> ### 2. On the Independence of H2 and H3 (Pattern Inference vs. Execution):
>
> The reviewer is correct; our phrasing "validated over" was imprecise, as the hypotheses are independent. We will revise the paper to clarify our finding: **pattern inference (H2) appears to be a more significant bottleneck for models than pattern execution (H3)**. The difficulty in inducing the correct rule seems to be the primary failure point.
>
>
> ### 3. On the Fairness of LRM Comparisons:
>
> We agree that the initial LRM comparison lacked ideal controls. To address this, **we have run new experiments with well-controlled model pairs**, where each pair shares the same architecture and pre-training (Deepseek-v3/R1 and **Qwen-2.5-32B/QwQ-32B**).
>
> The new results below strengthen our original conclusion: reasoning-specialized models do not consistently overcome the "Curse of CoT" on these implicit pattern tasks.
>
> | Model           | MiniARC | COGS   | RAVEN  | Average |
> | :-------------- | :------ | :----- | :----- | :------ |
> | Qwen-2.5-32b    | 24.16   | 21.80  | 12.07  | 19.34   |
> | QwQ-32b         | 18.70   | 13.00  | 8.82   | 13.51   |
> | Deepseek-v3     | 27.52   | 30.80  | 21.05  | 26.46   |
> | Deepseek-R1     | 28.86   | 24.00  | 27.56  | 26.81   |
>
> ### 4. On Anthropomorphism in Language:
>
> This is a valuable suggestion. We will revise the manuscript to replace terms like "cognitive load" with more precise, technical descriptions of the model's processes.
>
> ---
>
> We thank the reviewer again for their time and feedback, which has helped us significantly improve the paper.
>
> **References**
>
> [1] Lost in the Middle: How Language Models Use Long Contexts. 2023.
>
> [2] RULER: What's the Real Context Size of Your Long-Context Language Models? 2024.
>
> [3] Multilingual Needle in a Haystack: Investigating Long-Context Behavior of Multilingual Large Language Models, 2024

---

> > ### Comment · Reviewer_2yea · 2025-10-01
> >
> > Thank you for the response. Points 2 and 4 would resolve my issues.
> >
> > Before I create my recommendation, I have one quick follow-up questions regarding point 1:
> >
> > 1. Could you please clarify if the rationale during the front-loading is the same for all data points (i.e., whether the rationale matches the data point) or if a rationale is once generated for a problem class (i.e., data set) and then re-used for the whole experiment?
> >
> > Further, I strongly encourage the authors to update the manuscript already, as this would make directly accepting the paper without requiring a revision easier.

---

> > > ### Author Response · Authors · 2025-10-01
> > > **Thanks for following-up**
> > >
> > > Thank you for the follow-up. Glad to know that our response addresses your concerns.
> > >
> > > ---
> > >
> > > ### 1. Clarification on Rationale Front-loading Experiment
> > >
> > > - The front‑loaded rationale is **matched per data point**. For each test instance, we first run the same LLM on the same instance under the regular CoT setting to elicit its rationale.
> > > - We then reuse that exact rationale (with the final answer stripped to avoid leakage) in the front‑loading condition, placing it before the demonstrations and query.
> > > - Thus, for each instance, the content of the rationale is identical across conditions; the only change is its position in the prompt, i.e., the contextual distance between demonstrations and the answer instruction.
> > >
> > >
> > > Here we would like to intuitively illustrate how our rationale front-loading experiment works:
> > >
> > > ---
> > >
> > > Direct Answering:
> > > `[demos] [question]` **`[answer]`**
> > >
> > > CoT:
> > > `[demos] [question]` **`[rationale] [answer]`**
> > >
> > > Rationale Front-loading:
> > > `[rationale] [demos] [question]` **`[answer]`**
> > >
> > > (`regular font`: provided in context; **`bold font`**: generated by LLM)
> > >
> > > ---
> > >
> > > ### 2. Manuscript Update
> > >
> > > Sure, we will update our manuscript shortly. Thanks for the suggestion.

---

> > > > ### Comment · Reviewer_2yea · 2025-10-01
> > > >
> > > > Okay. I suggest to make that explicit in the manuscript as well.
> > > >
> > > > That resolves all my concerns. I will then wait with my (positive) recommendation for after you update the manuscript.

---

> > > > > ### Author Response · Authors · 2025-10-01
> > > > >
> > > > > Sure. We have uploaded our revised manuscript accordingly.

---

### Review · Reviewer_2nPd · 2025-09-13

**Summary Of Contributions:**

The paper challenges the conventional wisdom that chain-of-thought (CoT) prompting “generally helps, especially for the pattern-based ICL settings (symbolic, textual tasks). The authors report that direct answering outperforms CoT/ReAct/ToT, and that the gap often widens with more demonstrations. To understand the mechanism behind the phenomenon, they propose various initial hypothesis including (i) a contextual-distance effect (rationales push the answer further away from the demonstrations, harming ICL),  (ii) LLMs struggle to infer underlying patterns (iii) LLMs struggle to apply underlying patterns. By designing and conducting the experiments, they finally synthesize their hypotheses to an explicit–implicit duality in CoT: explicit, verbalized pattern inference is often wrong, while an implicit mechanism (akin to direct answering) still sometimes recovers the right label. In last section, the paper also compare the SOTA LRM with SOTA LLM on these pattern-based tasks. Even long-CoT LRMs (o1-mini, DeepSeek-R1, QwQ-32B) fail to beat strong LLMs with direct answering despite far higher token costs, which further confirms the author's conclusion.

**Additional Comments:**

This paper is very insightful but it would be great (as mentioned above) to connect the phenomenon to the root cause we may improve during training.

**Audience:**

Yes

**Audience Explanation:**

This paper is well-motivated and propose a novel view to understand when and why the reasoning may help for LLM. This is an interesting and important topic in LLM domain.

**Broader Impact Concerns:**

I don't have any concerns on the ethical implication of this work.

**Claims And Evidence:**

Yes

**Claims Explanation:**

To understand the reason why CoT doesn't help in pattern-based tasks, the authors first give three hypotheses (contextual distance; difficulty of pattern inference; difficulty of pattern execution) and finally conclude a new dual-process view of CoT (explicit vs implicit) make the analysis structured and testable.

Specifically, they conduct clean ablations: 1). Dummy-rationale insertion (Shakespeare/countdown) to vary distance without adding task-semantics. Performance degrades as distance grows.  2). Rationale front-loading (prepend the rationale before demos) preserves semantics but removes the distance—performance improves, supporting the distance hypothesis.

In experiments, they compare the results of 16 LLMs on 9 datasets. They also compare the recent reasoning model with vanilla LLMs, which makes the insights more convincing.

**Requested Changes:**

I appreciate the effort and all the experiments designed and conducted to help people understand why CoT doesn't always help for pattern-based tasks. However, it would be more helpful if the author can design experiments or at least discuss the root cause for 1) why the dual-process happens  2) why the dual process happens especially for these "pattern-based" tasks (this is a descriptive word instead of strictly defined task/term). These questions may lead to deeper understanding of the issue during training process and help LLM researchers to enhance their models.

For point 1, For example, one hypothesis here can be that during training (especially post-training), most pattern-based tasks (IFT-tasks) are trained in a naive manner instead of CoT SFT or RL. Can more CoT training data bridge the gap?

For point 2, How to define the pattern-based tasks - in the more difficult but also somehow pattern-based tasks (e.g. formal methods, math), the same phenomenon still exists? In this paper, we only see the tasks that align with the author's hypothesis but don't know what's the boundary of these "pattern-based" tasks.

---

> ### Author Response · Authors · 2025-09-27
>
> Thank you for your insightful review and positive feedback. We appreciate you recognizing our structured analysis and the convincing nature of our evidence. You have raised excellent points about the deeper implications of our work.
>
> *   **On the root cause of the dual-process:** We agree that instruction-tuning with more labeled CoT data can help solve pattern-based tasks. However, we cannot attribute this phenomenon solely to the data formats used in the training corpus. The effect persists on benchmarks that are guaranteed to be contamination-free, such as SALT (published after the models' knowledge cut-offs) and RAVEN (which uses an abstract representation not found in online datasets). This suggests the cause is more fundamental than the training data format alone.
>
> *   **On the boundary of "pattern-based" tasks:** We propose understanding this boundary from the perspective of **pattern induction**. The datasets in our experiments can be framed as tasks of inducing a single, unified pattern from a set of input-output pairs. Consequently, these tasks reward *implicit pattern execution*, where the model induces and applies a latent rule from the in-context examples. In contrast, tasks like advanced math demand *explicit, multi-step reasoning*. For such tasks, the implicit execution paradigm is insufficient, which is why CoT offers a significant advantage, even without specific in-distribution training data.
>
> Thank you again for your valuable suggestions, we will revise our paper's discussion and clarify these key distinctions.

---

### Decision · Action_Editor_x4Rp · 2025-10-28

**Recommendation:** Accept as is

**Audience:**

Yes

**Audience Explanation:**

The reviewers agree that the paper is timely, well-motivated, and likely to be of broad interest.

**Claims And Evidence:**

Yes

**Claims Explanation:**

The reviewers all agree that the paper offers well-designed ablations to support its claims.